# Extreme springs in Switzerland since 1763 in climate and phenological indices

Noemi Imfeld[1,2], Koen Hufkens[1,2], and Stefan Brönnimann[1,2]

[1]Oeschger Center for Climate Change Research, University of Bern, Switzerland
[2]Institute of Geography, University of Bern, Switzerland

**Correspondence:** Noemi Imfeld (noemi.imfeld@unibe.ch)

**Abstract.** Historical sources report manifold on hazardous past climate and weather events that had considerable impacts on society. Studying changes in the occurrence or mechanisms behind such events is, however, hampered by a lack of spatially and temporally complete weather data. Especially, the spring season has received less attention in comparison to summer and winter, but is nevertheless relevant since weather conditions in spring can delay vegetation and create substantial damage due to late frost events. For Switzerland, we created a daily high-resolution (1x1 km$^2$) reconstruction of temperature and precipitation fields from 1763 to 1960, that forms together with present-day meteorological fields a 258-year-long gridded data set. With this data set, we study changes in long-term climate and historical weather events based on climate and phenological indices focusing on the spring season.

Climate and phenological indices show few changes in the mean during the first 200 years compared to the most recent period from 1991 to 2020, where climate change signals clearly emerged in many indices. We evaluate the climate and phenological indices for three cases of extreme spring weather conditions, an unusually warm spring, two late frost events, and three cold springs. Warm springs are much more frequent in the 21st century, but also in 1862 a very warm and early spring occurred. Spring temperatures, however, do not agree on how anomalously warm the spring was when comparing the Swiss temperature reconstruction with reanalyses that extend back to 1868. The three springs of 1785, 1837, and 1853, were particularly cold with historical sources reporting for example prolonged lake freezing and abundant snowfall. Whereas the springs of 1837 and 1853 were characterized by cold and wet conditions, in the spring of 1785 wet days were below average and frost days reached an all-time maximum, in particular in the Swiss Plateau indicating inversion conditions. Such conditions are in line with a high occurrence of north-easterly and high pressure weather types and historical sources describing *Bise* conditions, a regional wind in the Alpine area related to inversions. Studying such historical events is valuable since similar atmospheric conditions can lead to cold springs affecting vegetation growth and agricultural production.

## 1 Introduction

Studies of long-term climate variability often focus on the summer or winter season. However, climate in spring is equally important, for example, for plant growth, and may have far reaching impacts. Cold spells in spring can delay crop growth considerably, late frost events can destroy subsequent harvests (Vitasse and Rebetez, 2018), and spring snowfall may put trees

at risk due to snowfall (e.g. Tavankar et al., 2019; Schelhaas et al., 2003). Moreover, late beech leaf unfolding may prolong the spring wildfire season as sunlight penetrates to the ground and dries the litter layer (Valese et al., 2011). Studying adverse weather conditions in spring requires daily data, from which targeted climate indices can be calculated. Studies have evaluated changes in climate indices over the last few decades (e.g. Brown et al., 2010; Domínguez-Castro et al., 2020; Zhang et al., 2011), also focusing specifically on spring conditions, such as changes in late frost occurrence, safety margins of plants, and false springs (e.g. Wypych et al., 2017; Vitasse et al., 2018; Zahradníček et al., 2023). However, only very few studies have extended analyses of daily-based climate indices over several centuries because the necessary temporally complete daily data are rarely available (Brugnara et al., 2022; Diodato et al., 2020; Parker et al., 1992). In particular from a historical perspective evaluating daily-based indices, such as the occurrence of frost days or the occurrence of the last frost day in spring can be relevant as they are often reported in historical documents (Zhang et al., 2011; Pfister et al., 2017). Studying historical weather conditions in spring may also contribute to our understanding of today's adverse spring weather since it extends the sample of extreme events.

For Switzerland, we created a daily high-resolution (1x1 km$^2$) reconstruction of temperature and precipitation fields from 1763 to 1960 (Imfeld et al., 2023), that forms together with present-day meteorological fields a 258-year-long gridded data (MeteoSwiss, 2021a, b). This data set allows us to calculate impact-relevant climate and phenological indices for the 258-year-long period and to study long-term spring climate and past extreme springs since 1763. A dense network of phenological observations exists in Switzerland starting in 1951 (Auchmann et al., 2018; Brugnara et al., 2020a), but for earlier periods, historical phenological observations are sparse. Thus, we used numerical approaches to model phenology from the gridded daily meteorological data.

This article is organized as follows. In Sect. 2, we describe the meteorological data used to calculate climate and phenological indices. In Sect. 3, we describe the calculation of the climate indices and the phenological application. In Sect. 4, we describe the long-term changes in the climate and phenological indices and analyze these indices for three different extreme spring cases, a warm spring, two frost events, and three cold springs. These results are discussed in Sect. 5. In Sect. 6, we conclude our article.

## 2 Data

### 2.1 Meteorological data

For the calculation of climate and phenological indices, we used a reconstruction of 258 years of daily mean temperature and daily precipitation sums for Switzerland covering a period from 2 January 1763 to 31 December 2020 with a resolution of 1 km (Imfeld et al., 2023). For precipitation, the gridded data set also covers catchment areas outside Switzerland. These meteorological fields were reconstructed with the analog resampling method, quantile mapping, and data assimilation. The analog resampling and data assimilation are performed using a large number of instrumental measurement series from Switzerland and neighbouring regions. The data sets consist of two main sub-periods with different reconstruction skills due to the availability and quality of the input data. From 1763 to 1864, the reconstruction shows good skills for daily temperature with

correlations on average between 0.80 and 0.96 (calculated from the anomalies of a climatological annual cycle) and root mean squared errors on average between 1 and 2.6 °C depending on the input station network and season. For precipitation, the reconstruction skill for the period from 1763 to 1864 is lower with correlations on average between 0.6 and 0.8 and root mean squared errors between 6 and 10 mm depending on station network and season. However, the number of monthly wet days (daily precipitation $\geq$ 1 mm) compares well with an independent series of wet days from Bern as shown in Imfeld et al. (2023). After 1864, reconstruction skills are much improved across Switzerland for both temperature and precipitation data. Despite the drawbacks in the early period, we used this novel data set, since it is the first one offering daily data at a high spatial resolution for Switzerland.

To represent the uncertainty in the phenological indices, for temperature we used additionally an ensemble of the ten best analogue days with subsequent by data assimilation (see Imfeld et al., 2023). As the first analogue day is a better representation of a historical day than the second analogue day, this ensemble is not equivalent to a commonly used ensemble where all members are equally likely. A detailed description of the entire reconstruction and reconstruction method is found in Imfeld et al. (2023).

## 2.2  Reanalyses and weather types

In addition, we used the station-based daily temperature time series for the Swiss Plateau derived from the series of Bern and Zurich (Brugnara et al., 2022), further denoted as the "Swiss series" and the reanalysis 20CRv3 starting in 1807 (Slivinski et al., 2019) to calculate the same indices. For monthly mean values, we used the Modern Era Reanalysis ModE-RA (Valler et al., 2024). Note that these data sets are not fully independent of the Swiss reconstruction since they all rely partly on the same input data. From the reanalysis 20CRv3, we selected out of the four closest grid cells, the cell that correlates well and had a low bias compared to the Swiss Plateau area mean value. For ModE-RA, we used the grid cell in the northwest of Switzerland because it had low biases in comparison to the Swiss Plateau area mean. 20CRv3 assimilates pressure observations from Hohenpeissenberg, Torino, and Geneva, which are also used in the Swiss reconstruction (Imfeld et al., 2023). ModE-RA assimilates pressure, temperature, and wet days, but on a monthly resolution. 20CRv3 and ModE-RA were further used to analyze atmospheric conditions during the extreme spring examples. To evaluate the occurrence of weather types, we considered the reconstruction of Schwander et al. (2017) starting in 1763.

## 3  Methods

### 3.1  Climate indices

We selected nine different climate indices for daily mean temperature and precipitation (Table 1, upper part) which have been suggested by the Expert Team on Climate Change Detection and Indices (ETCCDI) (e.g. Zhang et al., 2011). They are not exclusively based on spring weather, i.e. some are influenced already by winter temperatures, but they generally relate to conditions in spring. Because only daily mean temperature is available for the period since 1763, we adjusted the indices to

daily mean temperature. A frost day was defined as a day with a daily mean temperature equal to or below 0 °C. Such a day is

thus colder than what is normally considered a frost day, but it is warmer than an ice day, for which maximum temperature needs to be below 0 °C. The warm and cold spell indices were calculated for the 10th and 90th percentile thresholds in the reference period of 1871 to 1900 for daily mean temperature and not minimum and maximum temperature. Further, we calculated the growing season start based on the first six days of the year with daily mean temperature above 5 °C. The growing season length index, which is based on the growing season start, has been criticized for a high inter-annual variability related to the fact that

the index operates on synoptic time scales (Cornes et al., 2019), rather than representing the conditions within a whole season. Thus, we also used the growing degree days index (GDD) that can be seen as a starting point for spring vegetation based on different thresholds (Wypych et al., 2017). Here, we used a threshold of 200 growing degree days.

For precipitation, we only focused on wet days related indices since these are better represented in the reconstruction. Thus, we calculated the seasonal wet days, consecutive wet days, i.e. the maximum number of consecutive wet days in a season, and

an estimate of snowfall days. Snowfall days were estimated according to Zubler et al. (2014) using a threshold of at least 1 mm for precipitation and less than 2 °C for daily mean temperature. We evaluated these thresholds by comparing the closest grid cells of snowfall days with observations of snowfall days at 27 different stations across Switzerland for the spring season. The results showed a mean Spearman correlation of 0.8 and a mean bias of -0.3 d. Especially for stations above 1000 m, the gridded data set showed a tendency to underestimate the number of snowfall days. It is therefore important to remember that

our snowfall days only represent potential snowfall days. Nevertheless, these estimates provide a good basis for evaluating spring snowfall over the historical period.

All indices were calculated for the entire available period from 1763 to 2020 and for the spring months from March to May for aggregated indices. Long-term changes in the indices were discussed based on 30-year mean values for eight climatological periods from 1781 to 1810, 1811 to 1840, 1841 to 1870, 1871 to 1900, 1901 to 1930, 1931 to 1960, 1961 to 1990, and 1991

to 2020. Anomalies are shown as a deviation from the period from 1871 to 1900. This period is defined as the pre-industrial reference period by the Swiss weather service based on comparison with other periods (Begert et al., 2019). Therefore, we used this period as a reference period in the herein presented results. For the calculation of indices based on the Swiss series (Brugnara et al., 2022), the entire year/season was set to missing for the indices growing season start and growing degree days if a missing value occurred. For aggregated indices (e.g. frost days, wet days), a value was set to missing if more than 10 %

of the values in an aggregation period were missing. All indices and their definitions can be found in Table 1. All calculated indices for a monthly, seasonal, and annual time aggregation for the period 1763 to 2020 are published at the open-access repository PANGAEA (Imfeld et al., 2024).

### 3.2 Phenological application

To study the impacts of past weather on the spring vegetation, we calculated the cherry full flowering and beech leaf unfolding

120  day-of-year from daily mean temperature data. Cherry flowering occurs around mid-April in the Swiss Plateau and is therefore a good indicator of the state of the spring vegetation. Beech leaf unfolding occurs around the beginning of May in the Swiss

**Table 1.** Climate and phenological indices. The phenological model are further described in Sect. 3.2.

| Climate index | Definition | Units |
|---|---|---|
| Growing season start | First day of at least 6 days with daily mean temperature > 5 °C | day-of-year |
| Growing degree days | Accumulated temperature > 5 °C reaching 200 GDD | day-of-year |
| Frost days | Number of frost days with daily mean temperature $\leq$ 0 °C | days |
| Last frost day | Last day of the first half of the year with daily mean temperature $\leq$ 0 °C | day-of-year |
| Cold spell index | Number of 5 consecutive days with daily mean temperature > 10th percentile | days |
| Warm spell index | Number of 5 consecutive days with daily mean temperature < 90th percentile | days |
| Wet days | Number of days with daily precipitation $\geq$ 1 mm | days |
| Consecutive dry day | Maximum number of consecutive days with daily precipitation < 1 mm | days |
| Snowfall days estimate | Number of days with daily mean temperature < 2 °C and daily precipitation sum $\geq$ 1 mm | days |
| **Phenological index** | **Description and scientific name** | **Model** |
| Cherry full flowering | Prunus avium - flowering (50 %) | PTT |
| Beech leaf unfolding | Fagus sylvatica - leaf unfolding (50 %) | TT |
| Frost index | Accumulated daily mean temperature $\leq$ 0 °C after cherry full flowering | - |

Plateau and is thus representative of later spring vegetation. The phenological phases refer to the day of the year when 50 % of the cherry tree is blooming, resp. 50 % of the beech leaves are unfolded.

For calibrating and reconstructing the phenological phases during the 258-year long period we used the phenological obser-
vations of the Swiss phenological network (SPN) between 1951 and 2020. Only series with a quality class of at least 3 were used leading to a total of 68 (56) time series for cherry flowering (beech leaf unfolding) with record lengths between 35 and 71 years distributed across Switzerland (Auchmann et al., 2018; Brugnara et al., 2020a). These quality classes were defined based on the length of records, completeness (missing values and number of gaps >5 years), reliability (number of quality flagged values), and number of inhomogeneities in the record. For further details on the quality classes see Auchmann et al. (2018). For
the cherry full flowering only grid cells below 1600 m a.s.l. were considered because the observational network only includes observations below this altitude.

The cherry full flowering dates were estimated using a photo thermal time model (PTT) as implemented in the phenor R package by Hufkens et al. (2018). It is based on the growing degree days temperature response ($R_g$) and a term accounting for the photoperiod estimated based on the daylength $L_i$ of day i (Eq. 1 and 2 ). Temperatures are accumulated starting at day $t_0$
until day n, if the daily mean temperature is above the threshold $T_{base}$. $T_i$ represents the daily mean temperature of day i. When the state of forcing, $S_{frc}$, reaches a specific threshold $F_{crit}$, the phenological phase happens. The parameters $F_{crit}$, $T_{base}$, and $t_0$ are calibrated based on the phenological observations from 1951 to 2020.

$$R_g(T_i) = \begin{cases} 0 & \text{if } T_i \leq T_{base} \\ T_i - T_{base} & \text{if } T_i > T_{base} \end{cases} \tag{1}$$

$$S_{frc} = \sum_{i=t_0}^{n} \frac{L_i}{24} R_g \geq F_{crit} \tag{2}$$

The beech leaf unfolding dates were estimated based on the thermal time model. This model is based on the growing degree days temperature response $R_g$ (1), but the daylength is not considered (3).

$$S_{frc} = \sum_{i=t_0}^{n} R_g \geq F_{crit} \tag{3}$$

     Both models are solely based on a forcing response of temperature and do not include chilling accumulation, i.e. the accumulation of daily mean temperature below a certain threshold. The evaluation of all models based on daily mean temperature

and included in the phenor R package (Hufkens et al., 2018) showed rather similar evaluation results with the two models selected here performing slightly better. The lack of winter chilling in our models can, however, influence the estimated phenological phases. The two evaluated species both have a negative correlation between the chilling accumulation and the heat requirement for their spring phenology (Wang et al., 2020). Calibrating a model on observations already affected by climate change (especially the years between 1990 and 2020) without considering winter chilling might, thus, lead to a higher heat

requirement as shown in Wang et al. (2020). Transferring these calibrated models then to the past could lead to phenological dates being too late in spring. A comprehensive study of these effects would be needed but was beyond the scope of this article.

     The model parameters were calibrated with a Markov Chain Monte Carlo differential evolution sampler with snooker update (Hartig et al., 2023; Ter Braak and Vrugt, 2008) and run with 18,000 iterations across 3 chains. Bayesian model calibration has been shown to perform well for the calibration of phenological models (e.g. Fu et al., 2012; Meier and Bigler, 2023) and it

further allows for assessing the uncertainty and convergence of model parameters. For the prior distributions of the parameters, a uniform distribution with pre-set bounds as defined in Hufkens et al. (2018) was used. The model calibration converged with a potential scale reduction factor of 1.02 or below (Gelman and Rubin, 1992). Figure A1 in the Appendix shows the trace of the calibrated parameters for the 6000 iterations and the marginal densities thereof. Figure A2b shows the root mean square error from a cross-validation based on station data. We added both phenological phases to the provided indices on PANGAEA

(Imfeld et al., 2024), but in the following only discuss cherry full flowering.

     For comparing the phenological reconstructions to independent historical observations, we used the time series for full flowering of cherry in Liestal (Canton of Baselland) starting in 1894 (Defila and Clot, 2001) and a composite time series of cherry flowering from different historical sources representative of the Swiss Plateau (Burgdorf et al., 2023; Rutishauser et al., 2003).

The cherry full flowering was further used to study the frost occurrence after flowering that could potentially cause damage to trees. A frost index was calculated following Lhotka and Brönnimann (2020) by accumulating daily mean temperature below

0 °C from the onset of cherry flowering (minus 3 d) until 30 June. This yielded an estimate of the area affected by late frost and of the intensity of the frost occurring. Most current studies of spring frost events (Vitasse and Rebetez, 2018; Vitasse et al., 2018, e.g.) relied on daily minimum temperature which has shown different rates of changing compared to the daily maximum or daily mean temperature (Scherrer and Begert, 2019). Since minimum temperature is not available in the past, we did not look at changes over time but focused on the representation of specific historical frost events.

## 4 Results

### 4.1 Longterm changes in climate and phenological indices

Most climate indices showed few differences in their climatological mean in the first five periods from 1781 to 1900 (Fig. 1 and 2). The Swiss climatological reference period from 1871 to 1900 showed slightly colder conditions in some of the indices compared to the earlier periods. Growing degree days were up to five days earlier in the Swiss Plateau in the periods between 1781 to 1870 compared to the 1871 to 1900 period, up to four frost days less were registered in these periods, and up to eight warm spell days more. Figure A3 and A4 in the Appendix shows the anomalies of the seven periods with respect to the reference period from 1871 to 1900.

Warmer conditions emerged in all indices for the three periods from 1931 onward. The 1931 to 1960 period showed earlier growing season start and earlier growing degree days compared to the period from 1961 to 1990, but few differences in frost days, last frost days, and warm and cold spells. Very clear differences emerged in the last period of 1991 to 2020 (Fig. 1 and 2, last row). Across Switzerland, the growing season started up to 24 d earlier than in the reference period 1871 to 1900. The 200 GDD was reached up to 25 d earlier, and the last frost days occurred up to 25 d earlier compared to the reference period 1871 to 1900. Warm spell days increased by up to 20 d, whereas cold spell days decreased by up to 10 d. P values from the comparison of the climatological mean and the mean of the reference period with a Student's t test show that for most indices the last period significantly differs at a 0.05 confidence level (not shown).

Based on our reconstruction, cherry flowering occurred on average between mid and end of April in the Swiss Plateau during the reference period from 1871 to 1900 (Fig. 1 i). The flowering phase did not show considerable changes in the mean in the first five periods until 1930. Changes become apparent in the period of 1931 to 1960 at higher locations, and much more pronounced again in 1991 to 2020 with between 5 and 15 d earlier than in the reference period (see also in Appendix Fig. A3). The Student's t test showed significant differences at a level of 0.05 in parts of Switzerland already in the 1931 to 1960 period (not shown).

For the precipitation-related indices, differences between the periods were small (Fig. 2 and A4 c to e). The first two periods showed a north-south difference in the number of wet days compared to the mean of the reference period from 1871 to 1900. This is likely an artifact of the data set due to the lack of precipitation data before 1864 in southern Switzerland. The 1931 to 1960 period showed a lower estimate of snowfall days and wet days than in the reference period. This period also included a prolonged episode of warm and dry years between 1945 to 1952 in Switzerland and Western Europe (Imfeld et al., 2022). The period from 1961 to 1990 showed no differences in the snowfall days estimate compared to the reference period, whereas the

1991 to 2020 period showed up to five days fewer snowfall days. Similarly, wet days did not show any difference in the 1961 to 1990 period, whereas they decreased in the last period. For the snowfall days estimate, the differences in the mean value became significant at a 0.05 level for the last period compared to the reference period (not shown).

Time series for the area-mean of the Swiss Plateau region (Swiss recon), for the 20CRv3 reanalysis, the merged time series from Zurich and Bern (Swiss series), and the ensemble mean of ModE-RA depict a steep trend of the indices in the late 1980s for daily mean temperature and GDD (Fig. 3a and b). 20CRv3 shows lower temperatures, e.g. leading to later GDD, in the period from 1806 to around 1835, which is also a period where few observations were assimilated into 20CRv3 and considered the experimental extension (Slivinski et al., 2019, 2021). ModE-RA agrees well with the Swiss reconstruction and the Swiss series in the 18th and 19th centuries, whereas it is on average colder than the other data sets in the 20th century. Notably cold springs in the time series are 1785, which also showed a much higher number of frost days in the Swiss Plateau than any other year, and 1837, which is together with 1785 the coldest spring in the 258-year-long time series. On the other hand, several springs showed quite high temperatures, comparable to the warm springs of the 21st century. The most prominent among these is the spring of 1862. Less visible in the 30-year climatologies, but evident from the time series are also the periods in the late 1940s and around the 1860s which show overall higher spring temperatures and earlier GDD.

The trend towards earlier flowering is also seen in the cherry flowering time series, with considerable earlier dates after 1989 (Fig. 3d and e). For the cherry tree in Liestal, the Pearson correlation between the reconstruction and the observation was 0.85, but the reconstruction showed a mean bias of 7.36 d. The ten ensemble members of the temperature reconstruction are barely distinguishable from each other since differences between the individual members only range between -5 and +4 d. Overall, 94% of the estimated phenological dates differ only between -1 and +1 d between the members. The reconstruction uncertainty stemming from the temperature data is, therefore, small, and it is below the mean root mean squared error when calibrating the phenological model (see Fig. A2 for comparison). However, all members are based on the same temperature observations. Errors in these observations are not considered and could increase the uncertainty.

For the composite cherry flowering from Rutishauser et al. (2003), the Pearson correlation is 0.67, and the mean bias is 2.97 d considering the best reconstruction (Fig. 3e). As for Liestal, the ten ensemble members show very similar flowering dates. For the Swiss Plateau, on average the earliest flowering occurred in 2017 (4 April), which was related to a damaging frost event in Switzerland (Vitasse and Rebetez, 2018). Very early flowering events of the 18th and 19th centuries happened in 1815 (8 April), 1794 (10 April), and 1862 (11 April). The latest flowering happened in 1785 (15 May), followed by 1853 (12 May). Other late years with cherry flowering between 9 and 10 of May were for example 1770 (a prolonged cold and wet period; see Collet 2018 and Imfeld et al. 2023), 1817 (after the year without a summer; see Flückiger et al. 2017), 1808, 1932, and 1837.

Despite the biases between the reconstruction and the historical observations, the phenological reconstruction reproduced the overall variability throughout the years. Thus, the reconstruction offers an estimate to study cherry flowering in the past across Switzerland.

### 4.2 Examples of extreme springs

Based on these climate and phenological indices, we studied three examples of extreme spring conditions, that may affect vegetation growth in spring. Namely, we considered the early warm spring in 1862, the occurrence of late frost events in 1873 and 1957, and the three years 1785, 1837, and 1853 with especially cold springs and late cherry flowering. In addition to the presented indices, we analyzed atmospheric variables for illustration of the weather conditions during the extreme spring cases, and we qualitatively evaluated historical sources reporting the weather conditions and weather-related impacts.

### 4.2.1 The warm spring in 1862

Very warm springs considerably increased after the 1980s. However, also in the late 18th and early 19th century, several warm springs with high daily mean temperature and early reach of 200 GDD occurred (Fig. 3a and b). The spring of 1862 in particular, stands out with a mean temperature of 10.4 °C between March and May in the Swiss Plateau based on the gridded reconstruction. It ranks as the third warmest spring since 1763 after the two warmest springs 2011 (11 °C) and 2007 (10.9 °C). With respect to the climatological period from 1841 to 1870, 1862 was exceptionally warm with an anomaly of 2.9 °C for the Swiss Plateau area mean. In contrast, the second (1841) and third (1846) warmest springs in the 1841 to 1870 period showed less pronounced anomalies of 1.8 and 1.1 °C. The warmest spring in 2011 had an anomaly of only 1.8°C with respect to its mean climate from 1991 to 2020. We also considered the Swiss series of Bern and Zurich (Brugnara et al., 2022) and 2 m temperature from 20CRv3 (Slivinski et al., 2019). In the Swiss series, the spring of 1862 ranks seventh with a mean temperature of 10.1 °C between March and May. It had an anomaly of 2.8 °C considering the mean of the period from 1841 to 1870, whereas 2011 it had an anomaly of 1.8 °C considering its mean climate from 1991 to 2020. Thus, the anomalies were very similar and the spring of 1862 seemed to have been unusually warm for its period. In 20CRv3, the anomaly of the spring 1862 was lower with 1.9 °C with respect to the 1841 to 1870 period. Across all years, the spring of 1862 only ranks 17th in 20CRv3. In contrast, in ModE-RA, which ends in 2008, the spring of 1862 showed the highest temperature across the period from 1763 to 2008 and had an anomaly of 2.6 °C concerning the 1841 to 1870 mean. The spring of 2007 was the second warmest, but 2011 is missing for comparison. The number of assimilated observations in ModE-RA, however, gradually reduces towards the 21st century affecting the temperature analysis (Valler et al., 2024).

For both, the warmest spring of 2011 and the warm spring of 1862, climate indices showed above-average temperatures and an above-average number of warm spell days across the entire Switzerland, though much more pronounced in 2011 (Fig. 4). Both springs showed mostly fewer wet days and fewer snowfall days than on average between 1871 and 1900. Cherry flowering was in certain areas up to 24 d advanced, in particular at higher altitudes.

The reanalysis 20CRv3 showed only a very weak positive geopotential height anomaly at the 500hPa level over eastern Europe for the mean of the spring months March to May 1862 (Fig. 5a). For the spring of 2011, a more pronounced positive geopotential height anomaly at the 500hPa level was present over western Europe indicating that Switzerland was affected by the warmer and drier conditions (Fig. 5b). In the spring of 1862, a cold anomaly in mid-April interrupted the warm weather (Fig. 5c) leading to frost and snowfall over Switzerland, but, for example, for Aarau as mentioned above no reports on vege-

tation damage were found (Zschokke, 1865). After the cold spell, a pronounced ridge established again over western Europe continuing to the warm spring weather (Fig. 5d). In ModE-RA, for which 1862 is the warmest spring in the 1763 to 2008 period, the geopotential height field at the 500 hPa level in spring was comparable to 20CRv3 with no pronounced ridge. But, the respective temperature anomalies were more pronounced than in 20CRv3 (Appendix, Fig. A5).

Historical sources indeed reported an unusual early snow-free period in spring 1862 in Ursern, a valley in the Canton of Uri (see Table 2 and Fig. A6). Already very early in the year the Gotthard was passed by carriage and not sled (Zschokke, 1865) which points to warm weather leading to early snow-melting, but also to less snowfall in the months before. For Aarau, Theodor Zschokke reported unusual advances in the vegetation, for example, a start of the cherry flowering as early as the 6th of April. In our reconstruction the cherry flowering happened on the 8 April in Aarau and on average on the 11 April in the Swiss Plateau. The snowfall and frost that occurred in mid-April did not lead to damage in lower-lying areas (Zschokke, 1865). An official weather report from the weather service for the year 1862, however, did not mention an unusually warm spring (MeteoSwiss, 2016), but more qualitative sources describing the spring weather might be available.

### 4.2.2 The late frost events in 1873 and 1957

Combining the cherry flowering reconstruction with climate indices allows us to look at climate events that affected vegetation directly, such as the occurrence of late frost in spring that can lead to considerable damage to vegetation. Two events stand out when considering the affected area and the intensity of the frost events (Fig. 6). In 1873, frost conditions after the cherry flowering affected large parts of Switzerland, however, the daily mean temperature did not fall much below 0 °C. The frost index based on the accumulated negative temperature reached at most -6 °C in the Swiss Plateau. The last frost day occurred between the 26th and 28th of April across northern Switzerland, which is more than half a month later than it occurred on average between 1871 and 1900. In contrast, cherry trees reached their full bloom up to 10 d early (Fig. 6b). In the spring of 1957, a frost event occurred with very low temperatures, but the affected area was smaller (Fig. 6a). The frost index showed much higher values but affected only areas above 800 m a.s.l. For these areas, the last frost days, which occurred between the 6th and 8th of May, were almost a month later than between 1871 and 1900 (Fig. 6c), and the cherry tree flowering was considerably earlier.

For both springs, March was characterized by average or above-average temperature conditions across Central Europe, though more pronounced for March 1957 (Fig. 6d). These high temperatures in March likely led to an early start of the cherry flowering. In March 1873, temperature anomalies were positive, but the geopotential field shows a more zonal configuration (Fig. 7). On the 26th of April 1873, a large trough extended over Switzerland from the Northeast leading to the temperature drop. In 1957, temperatures reached their lowest values on the 8th May when a large trough was located above Switzerland.

For both events, damage caused by the late frost events was reported. In 1873, frost damage was reported for Sursee, Marschlins, Bad-Ragaz, and Appenzell-Innerrhoden, whereas many locations registered snowfall during the 26th to 28th April leading to further damage to the vegetation (Table 2 and Fig. A6). For the frost event of 1957, the Swiss farmer association calculated a reduction of yield in pear and apple trees of 75 % and for cherries of 44 % compared to the six preceding years indicating considerable loss in harvests (Table 2).

**Table 2.** Selection of weather impacts for the extreme spring examples based on the Euro-Climhist database (Pfister et al., 2017) and further sources. The original sources are listed where possible. Ct. refers to the cantons (member states) of Switzerland. Numbers are marked on the map of Switzerland in the Appendix Figure A6.

| Time | Location | Impacts | Sources |
|------|----------|---------|---------|
| **Warm springs** | | | |
| 1862 Spring | Ursern and Gotthard (Ct. of Uri)[1] | unusual early snow-free period | Ambühl (1961); Zschokke (1865) |
| 1862 April | Aarau (Ct. of Aargau)[2] | unusual advances in vegetation | Zschokke (1865) |
| **Frost events** | | | |
| 1873 April | Sursee[3] | complete damage of cherry harvest | MZA (1873) |
| 1873 April | Marschlins[4] | damage of walnut trees and grapes | MZA (1873) |
| 1873 April | Switzerland | snow/frost damage on fruit trees and grapes | Appenzeller-Kalender (1874) |
| 1957 April/May | Switzerland | considerable losses in fruit harvest | SBV (1958) |
| **Cold springs** | | | |
| 1785 Jan-Apr | Lake Constance[5] | water bodies frozen | Paffrath (1915) |
| 1785 March | Lucerne[6] | water bodies frozen | Staatsarchiv-Lucerne (1755-1829) |
| 1785 March | Geneva[7] | water bodies frozen, ice partially walkable | Forel (1892) |
| 1785 March | Lake Constance[5] | snowfall | Paffrath (1915) |
| 1785 March | Thur river (Ct. of St Gall)[8] | water bodies frozen | Braecker (1998) |
| 1785 March | Lake Alpnach[9] | water bodies frozen | Schaller-Donauer (1937) |
| 1785 March | Lake Zurich[10] | water bodies frozen | Müller (1878) |
| 1785 March | Wattwil (Ct. of St Gall)[11] | several reports on abundant snow | Braecker (1998) |
| 1785 March | Chur (Ct. of Grisons)[12] | reports on abundant snow, 9 d snowfall | Grimmer (2019) |
| 1785 March | Lake Constance[5] | reports on abundant snowfall | Paffrath (1915) |
| 1785 March | Binn (Ct. of Valais)[13] | reports on birds frozen to death | Zennhäuser (2008) |
| 1785 April | Lake Constance[5] | snow impact, abundant snow, livestock starved | Paffrath (1915) |
| 1785 April | Chur (Ct. of Grisons)[12] | reports on abundant snow, 6d snowfall | Grimmer (2019) |
| 1785 April | Binn (Ct. of Valais)[13] | vegetation delayed, livestock starved | Zennhäuser (2008) |
| 1785 March | St Blaise (Ct. of Neuenburg)[14] | rigorous cold, abundant snow, and strong *Bise* | Kopp (1873) |
| 1785 April | St Blaise (Ct. of Neuenburg)[14] | abundant snow, cold, and strong *Bise* | Kopp (1873) |
| 1785 May | Lake Constance[5] | delayed vegetation and frost impact on vegetation | Paffrath (1915) |
| 1785 May | Wattwil (Ct. of St Gall)[11] | fresh snow at higher elevations | Braecker (1998) |
| 1837 March | Canton of Zurich[15] | snow and rain quantities 'as expected for a January' | Vogel (1841) |
| 1837 March | Simplon (Ct. of Valais)[16] | 13 death due to avalanche | Joller (1888) |
| 1837 April | Canton of Zurich[15] | large snow and rain quantities | Vogel (1841) |
| 1837 May | Canton of Zurich[15] | large snow and rain quantities | Vogel (1841) |
| 1837 May | Grindelwald (Ct. of Bern)[17] | permanent snow cover | Strasser (1890) |
| 1853 May | Canton of Zurich[17] | large snow and rain quantities | Vogel (1841) |
| 1853 May | Grindelwald (Ct. of Bern)[17] | permanent snow cover | Strasser (1890) |

### 4.2.3 The cold springs of 1785, 1837, and 1853

The three springs, 1785, 1837, and 1853 registered the lowest temperatures of the entire time series of 258 years and lie below the 1% quantile of all springs temperatures. Their mean spring temperature in the Swiss Plateau reached only between 4.1 to 5.1 °C (Fig. 3) which is up to 3 °C degrees colder than the 1871 to 1900 average (Fig. 8a). In the Swiss series, the coldest spring was in 1837 with a mean temperature of only 3.8 °C and an anomaly of -3.4 °C followed by 1785 with a mean temperature of 4.1 °C and 1853 with a mean temperature of 5.1 °C. In 20CRv3, the spring of 1837 ranks the coldest with an anomaly of -4.0

305 °C with respect to 1871 to 1900, but 1785 is not available. In ModE-RA, the coldest spring was registered in 1837, followed by 1785, 1970, and 1853 with anomalies between -2.2 and -1.8 °C with respect to 1871 to 1900.

Indices show, that during the three springs, the cherry flowering was up to 20 d later than the average of 1871 to 1900, and up to 30 more frost days were registered (Fig. 8b and c). Snowfall day anomalies were positive in the spring of 1837 and 1853 especially in the Alps, but they were negative in the Alps in the spring of 1785. The spring of 1785 also registered fewer wet

310 days than in the 1871 to 1900 period and thus did not concur with the two other springs, that showed wet and cold conditions. In 1785, the frost days indeed showed a different spatial pattern with much larger frost day anomalies in the Swiss Plateau region compared to the Alps, which would correspond to an inversion situation. This suggests that synoptic conditions were different over Europe during 1785 compared to the two other cold springs.

Weather types allow a look at the synoptic conditions throughout the three cold springs. The late springs of 1837 and

315 1853 show a higher occurrence of northerly cyclonic situations (N) and cyclonic situations with westerly flow over Southern Europe (WC) compared to usual spring weather types in the entire period from 1763 to 2020 (for the description of weather types see Schwander et al., 2017) (Fig. 9a). More cyclonic weather types are found also on average for all springs with temperatures below the 10th quantile (q10). The spring of 1785, however, showed an increase in weather types describing easterly, indifferent flow (E), and high-pressure situations over Europe (HP). To further evaluate this difference in weather

types, we used the variance of bandpass-filtered daily pressure observations which give an insight into the "storminess", i.e. the frequency of passing of extra-tropical cyclones. We followed the approach of Brugnara et al. (2015), but only considered station data for calculating anomalies of the standard deviation. The pressure observations show that storminess decreased for northern stations in 1785, and it increased in 1837 and 1853 over Switzerland (Fig. 9b). Also, the average of the springs below q10 shows increased storminess except for two stations.

This is confirmed by the monthly fields of geopotential height (anomalies) at the 500hPa level during the three spring months March to May (Fig. 10). In February and March 1785, a trough was present over Central Europe leading to advection of cold air from the north. This situation weakens through April and May, but the trough remains throughout spring. Stations from several locations in Germany, Poland, and the Czech Republic all registered the same extended negative temperature anomalies throughout March until mid April (not shown). In 1837 and 1853, more zonal flow patterns seemed to prevail between March

and May and the average of springs below q10 similarly shows more zonal flow.

Historical sources confirm these cold spring weather conditions. In spring 1785, for six lakes reports about continued (partial) lake freezing were found until March (see Table 2). In March and April, several locations reported abundant snowfall. Due to

feed shortage livestock starvation was reported from the Canton of Valais. In St Blaise in the Canton of Neuenburg, strong *Bise* was reported for March and April (Kopp, 1873). The *Bise* is a wind in the Alpine area channeled between the Jura mountains and the Alps and is related to high pressure and anticyclonic weather conditions (Wanner and Furger, 1990). It is further associated with the advection of cold and dry continental air, and the co-occurrence of stratus formation. Also, for St Blaise Kopp (1873) reported that the first rain (not snow) fell after 4 months at the end of May. Both, the *Bise* conditions and late rain indicate specific synoptic conditions for this cold spring. For the springs of 1837 and 1853, fewer sources are available, but they reported abundant rain, snowfall, and frost impacts for various locations in Switzerland. These sources, thus, confirm that all three springs were very cold.

## 5   Discussion

Climate and phenological indices provide a useful way to study historical extreme spring events, such as cold springs or late frost events, and relate them to impacts, for example, the state of vegetation. All indices showed steep changes towards warmer conditions in the last climatological period from 1991 to 2020. These changes in temperature indices correspond to the widely reported trends for temperature development in Switzerland such as in Isotta et al. (2019) for monthly means and in Xoplaki et al. (2005) considering several centuries and a European scale, as well as to changes of, for example, snowfall vs. rain (Serquet et al., 2011). The changes in temperatures are also seen in the advancement of spring phenology, which serves as a relevant bio-indicator for changing temperatures (Studer et al., 2005; Vitasse et al., 2018; Rutishauser et al., 2008; Luterbacher et al., 2007). The steep changes towards warmer conditions in the late 1980s have been found in a variety of time series across the world, including vegetation, temperature time series, and snow time series (Reid et al., 2016; Marty, 2008), however, mainly series linked to spring and winter conditions. Sippel et al. (2020) suggested that the changes, which are linked to the cold season temperature, stem from internal variability superimposed on a long-term warming trend.

For Switzerland, phenological models have been used to study for example past frost events (Vitasse et al., 2018) for different tree species and changes in future frost events for grapevines (Meier et al., 2018), but no attempts have been made to extend phenological predictions in space and time. The transferability of the phenological models may be limited in space (e.g. Basler, 2016), and also in time because the sensitivities of the calibrated parameters may not be constant over time, especially when going back until the 18th century (Rutishauser et al., 2007). Based on linear regression parameters, Rutishauser et al. (2008) showed that the sensitivity of spring phenology to spring temperature has shown phases of both increase and decrease over the last 300 years. The comparison of historical phenological observations with our reconstruction shows systematic biases of several days for the series of Liestal, but a high correlation. Thus, the reconstruction can depict the inter-annual variability of cherry flowering for Liestal despite the bias. For the composite cherry flowering series, the bias is small with below 3 d, but the correlation is also lower. However, both reconstructions agree on notable events such as the very early flowering in 1794.

The indices allow us to gain insights into historical springs with unusual weather conditions. The warm spring of 1862 exhibited warm temperatures across the entire Switzerland, which was exceptional for this period, however, it is less exceptional in comparison to the recent warm springs such as 2011 or 2007. Temperature anomalies were high in spring 1862 in the gridded

reconstruction, the Swiss series (Brugnara et al., 2022), and ModE-RA, but temperature anomalies were lower in 20CRv3. The three former data sets are all based on the same temperature series of Bern and Zurich. The geopotential height field in 20CRv3 does not point towards an exceptionally warm spring of 1862, which shows for the year 2011 quite pronounced positive 500 hPa geopotential height anomalies and a ridge over Europe, but not for the year 1862. Similarly, ModE-RA does not show very pronounced ridge conditions either in the spring months, but it does have high temperature anomalies. For Germany and Central Europe, Glaser and Riemann (2009) developed a documentary-based monthly index which reported for all three spring months above average, but not exceptionally warm conditions. Pfister and Wanner (2021), however, classified the spring of 1862 as category 3, which is the warmest category of this index representative of the Swiss Plateau. An evaluation of biases in the spring temperature might be needed to put the spring temperatures in further context.

Late frost events after a warm period in spring can be particularly damaging for example to fruit harvest. For Switzerland, studies have evaluated how the frost risk has changed over the last decades, and how it may be affected by climate change in the future (Lhotka and Brönnimann, 2020; Meier et al., 2018; Vitasse and Rebetez, 2018; Vitasse et al., 2018), however historical events and their extent have not been studied in detail. Two notable frost events occurred in the spring of 1873 and 1957. Harvest data for 1957 showed a considerable loss for apple, pear, and cherry trees (SBV, 1958), which also agrees with the frost index showing strong negative values. Vitasse et al. (2019) identified the spring frost of 1957, as one of the most severe springs in terms of frost risk between 1930 and 2016 based on tree-ring data from different forest species across Switzerland. Also for the Apennines in Italy, Tonelli et al. (2023) noted considerable effects on the tree ring growth in 1957 due to the late frost event. This is in agreement with the extension of the cold surge to southern Europe, as it is seen in the temperature anomalies and geopotential height field from 20CRv3 (Fig. 7b). For the year 1873, only qualitative descriptions of frost damage were available. Since the frost index is based on daily mean temperature and not minimum temperature, it may not have captured the extent of the frost event entirely. But, the data set allowed us to track past frost events based on phenology and frost days and relate them to historical descriptions of the events. This could be done as well for earlier events, one example is the late frost event in 1802.

Lastly, we considered the three springs of 1785, 1837, and 1853, that registered the lowest spring temperatures in the Swiss reconstruction. Climate indices showed differences between 1785 and the two springs of 1837 and 1853 with respect to the occurrence of frost days, snowfall days, and wet days. Weather types, the storminess calculation, and the ModE-RA confirmed these differences. The spring of 1785 was under the influence of a pronounced cold trough with a higher occurrence of easterly, high-pressure, and northerly weather types. This led to particular cold, but also dry conditions. The frost day anomalies in the Swiss Plateau indicated likely prolonged inversion, and historical sources describe extended periods of *Bise* which can also lead to inversions and fog in the Swiss Plateau. Daily surface pressure fields over Europe would be needed to study the cold conditions in late winter and early spring of 1785 in more detail. In 1837 and 1853, higher storminess was found in accordance with a more zonal flow and low-pressure systems passing leading to above-average wet conditions. For the very cold winter of 1785, six of the large lakes of Switzerland reported that the lakes were frozen until March, which would also be indicative of cold winters. As suggested by Franssen and Scherrer (2008) lake freezing could be reproduced based on negative growing degree days for further phenological comparison.

The spring of 1785 followed the cold years after the Laki eruption which occurred in Iceland in 1783 (Yiou et al., 2014; Zambri et al., 2019), but the very cold period end of winter and beginning of spring of 1785 has not been studied in detail. Also, the spring of 1837 followed a volcanic eruption, namely the Cosiguna in Nicaragua in January 1835. After this eruption, several cold years were evident in tree and frost rings from Europe (Longpré et al., 2014). However, Longpré et al. (2014), also stated that a cooling trend was already noted before the eruption. The spring of 1853 has not been related to cold conditions in previous literature, and for this spring the three data sets do not agree concerning the magnitude of the negative temperature anomalies. The monthly index for Germany and Central Europe based on documentary sources (Glaser and Riemann, 2009), however, assigned cold values to all three springs. In March 1785, the lowest index of -3 was reached, while April and May were classified as -2 and 0. In 1837, all spring months were classified as -2, and in 1853, the spring months were classified as -3/-2/-1 (March/April/May). Based on a statistical reconstruction of European temperature fields, Xoplaki et al. (2005) marks the spring of 1785 as the coldest spring in their data set. Also, the Pfister temperature index (see e.g. Pfister and Wanner, 2021) classified the three springs as very cold, with all three springs in the category -3. For the Czech Republic, Brázdil et al. (2024) found that the spring temperatures of 1785 together with 1740 were the coldest in the reconstructed temperature series of Dobrovolný et al. (2010). Also, Pappert et al. (2021) noted the particularly cold March of 1785 in the temperature observations of the Societas Meteorologica Palatina, with persistent below-zero temperatures for many locations across Europe. The cold temperatures across Europe fit well with the pronounced trough found in ModE-RA in March 1785 (see Fig. 10a). These cold spring conditions, in particular the unusually cold and dry conditions in 1785, thus provide a rather unique case of cold spring conditions.

## 6  Conclusions

Climate and phenological indices allow us to depict changes in spring weather and to study extreme springs since the mid 18th century. The 258-year-long time series for the different indices all showed few changes for the first 200 years compared to the steep increase toward warmer conditions in the most recent decades. Notable are a warm period in the late 1940s and a warm period around the 1860s. Some extreme spring events were, however, evident from the time series. Based on the different indices, we evaluated three cases of such extreme spring weather conditions since the mid 18th century. The spring of 1862 was exceptionally warm compared to its climatological mean and it still ranks among today's warmest springs, although this ranking is highly dependent on the data set. Upper-level atmospheric fields do not indicate similarly pronounced conditions in spring 1862, such as they do for the very warm spring of 2011. The combination of phenology and frost days allows us to evaluate past frost events that caused damage to vegetation. Whereas for the warm spring of 1862, no frost damage was reported despite a cold air outbreak, for the two cases of 1873 and 1957, the frost index shows the affected areas and historical reports confirm the significance of these events.

In the period from 1763 to 2020, three springs showed very cold mean temperatures of at most 5.1 °C. Whereas the springs of 1837 and 1853 showed cold and wet conditions, during spring 1785 fewer wet days than on average from 1871 to 1900, and, thus, dry conditions were registered. An evaluation of weather types, of a storminess index based on bandpass-filtered

pressure data, and ModE-RA showed that the 1785 spring was related to more high-pressure conditions and northeasterly flow over Europe, which brought cold air towards Switzerland. The high frost day amounts in the Swiss Plateau and reports about *Bise* further suggest a synoptic situation favorable for prolonged inversion and fog in the Swiss Plateau from March to April. In 1837 and 1853, the zonal flow and mainly cyclonic conditions led to cold but also wet springs. Both, the spring of 1785 and 1837 occurred after volcanic eruptions in the extratropics and the subtropics.

The climate and phenological indices allowed us to get insights into different historical extreme spring events. It was possible to relate these springs to impacts through historical sources and to evaluate the atmospheric conditions behind them by considering further data sets. Studying such past springs might also nowadays be interesting to better understand the causes of cold and warm springs. In addition, modelling phenological phases allowed us to relate the historical weather conditions in a straightforward manner to impacts on vegetation due to late frost events.

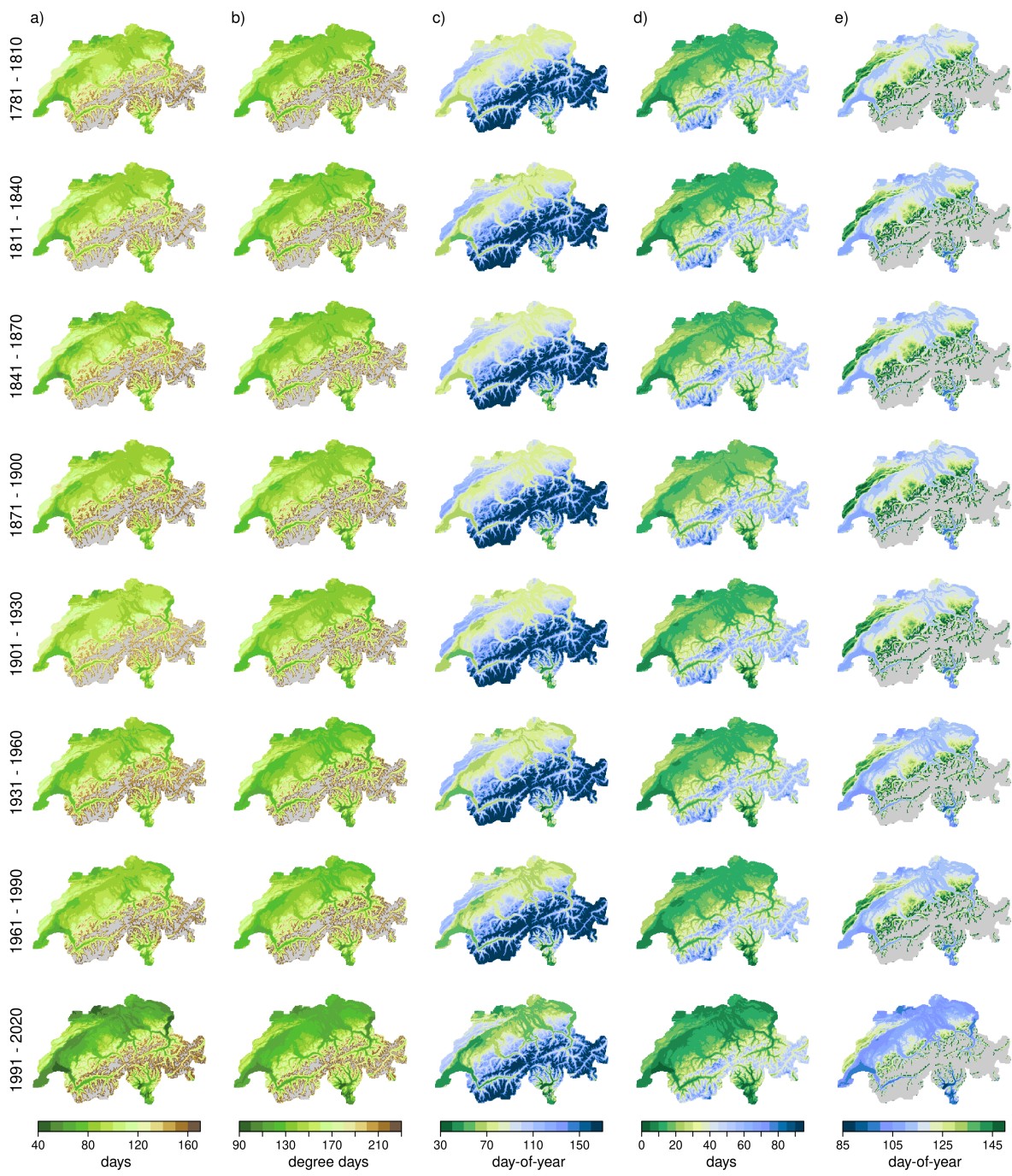

**Figure 1.** 30-year climatological mean for climate indices for the eight periods between 1781 to 2020. a) Growing season start, b) growing degree days, c) last frost day, d) frost days, and e) cherry full flowering day-of-year. Light grey areas depict areas, where the indices were not reached in more than 75 % of the years or we did not calculate the index because the grid cells are above 1600 m a.s.l. (last column).

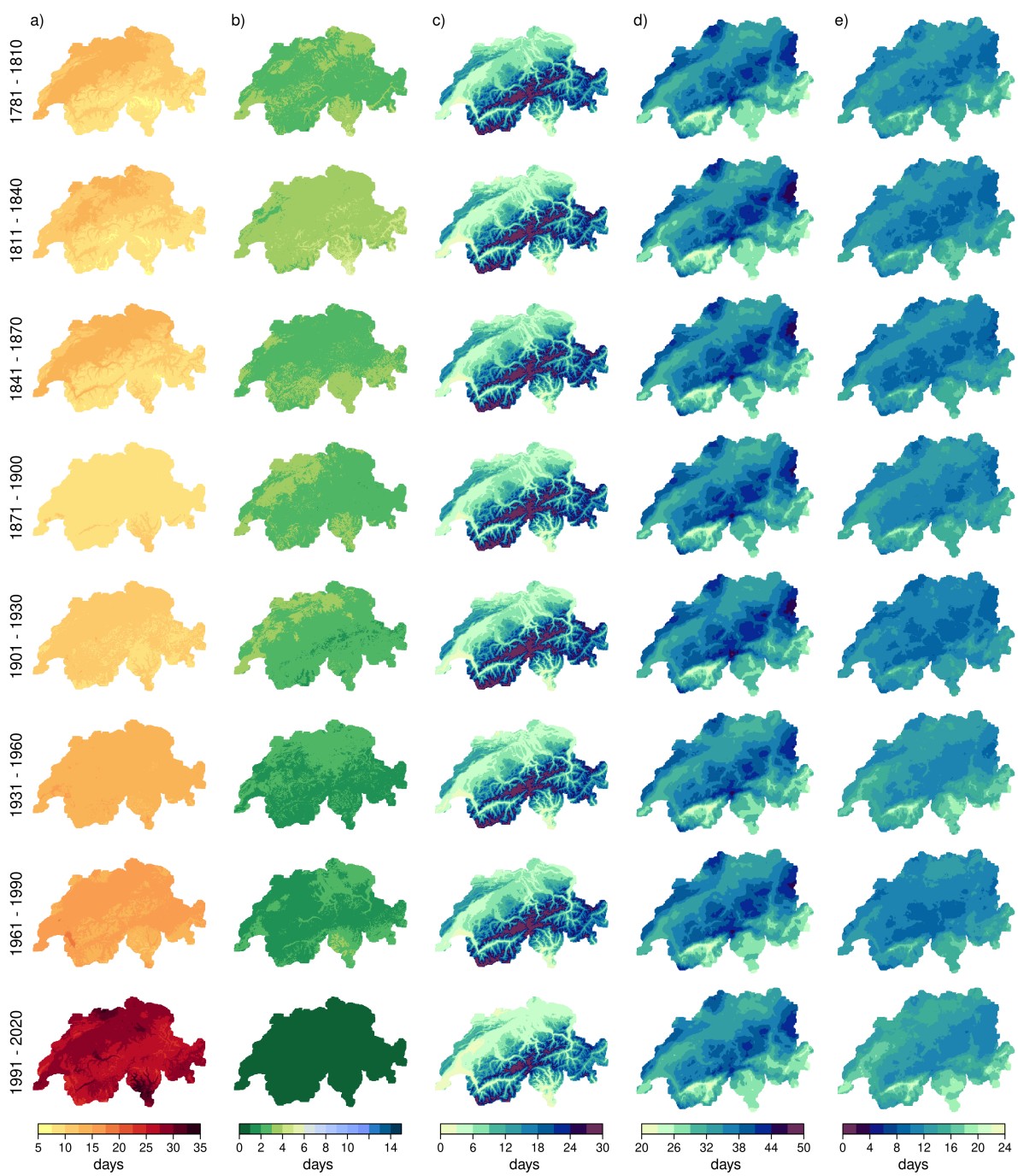

**Figure 2.** As in Figure 1, but for a) warm spell duration index, b) cold spell duration index, c) snowfall days, d) wet days, and e) consecutive dry days.

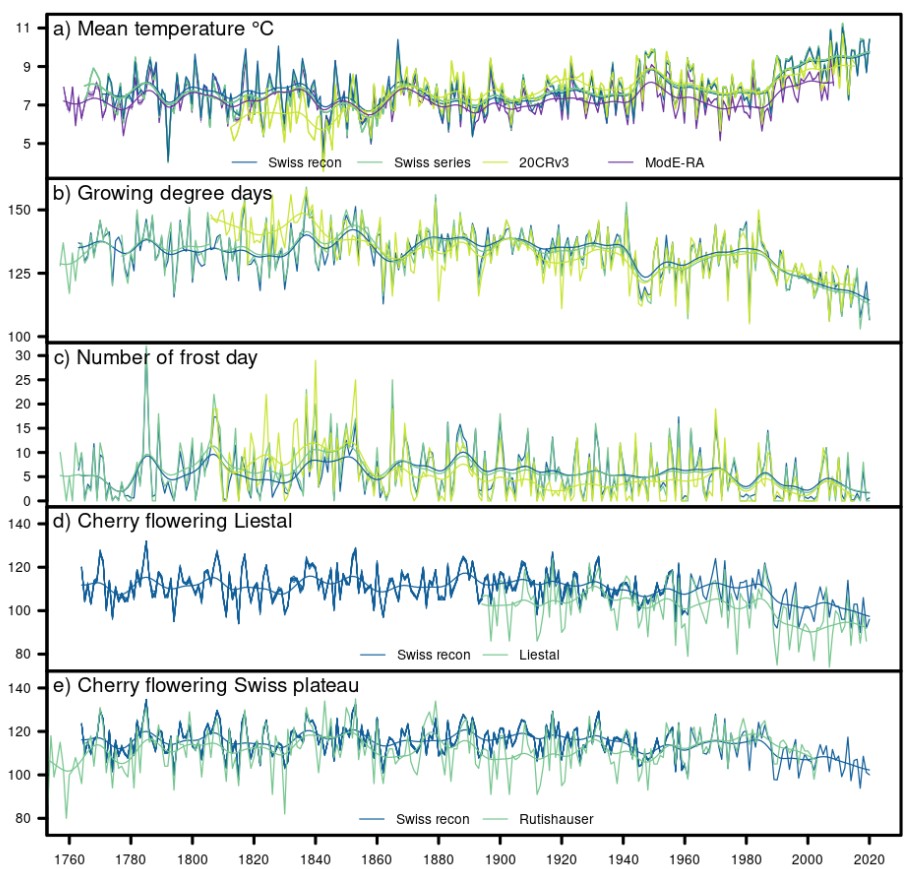

**Figure 3.** Evolution of indices in the spring season for different data sets. a) Mean spring temperature, b) growing degree days, c) number of frost days, d) cherry full flowering in Liestal for the reconstruction and the observations of Liestal (Defila and Clot, 2001), and e) mean cherry flowering in the Swiss Plateau for the reconstruction and the composite series of Rutishauser et al. (2003). Note that for the Swiss series, NA values are removed. The smoothed lines are produced with a Gaussian filter using $\sigma=3$ years. The time series of the Swiss gridded reconstruction covers the area of the Swiss Plateau. ModE-RA shows the ensemble mean, and the minimum and maximum member. 20CRv3 shows the ensemble mean and the spread. For the cherry full flowering reconstructions (d and e), we also show an ensemble of ten reconstructions from the analogue reconstruction.

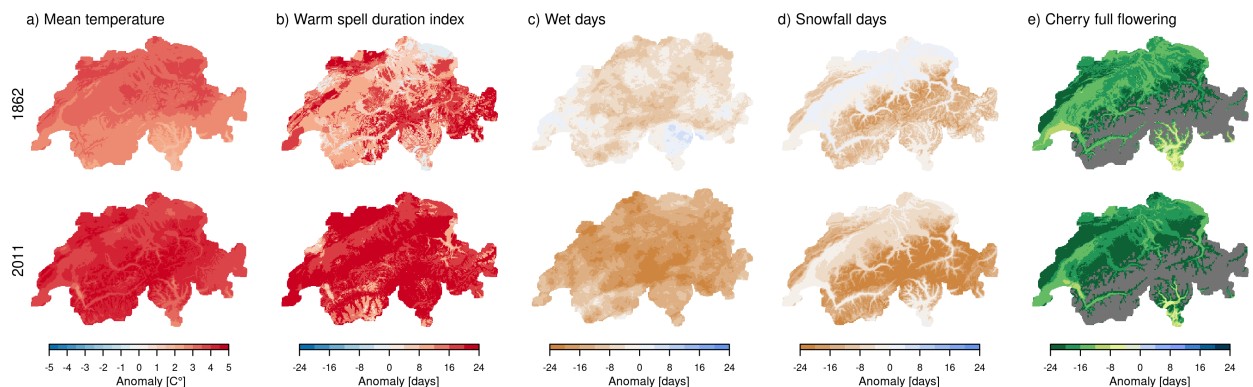

**Figure 4.** Anomalies of five indices for the warm springs of 1862 and 2011 with respect to the climatological mean values in the period from 1871 to 1900. a) mean spring temperature, b) warm spell duration index, c) number of wet days, d) number of estimated snowfall days, and e) cherry full flowering.

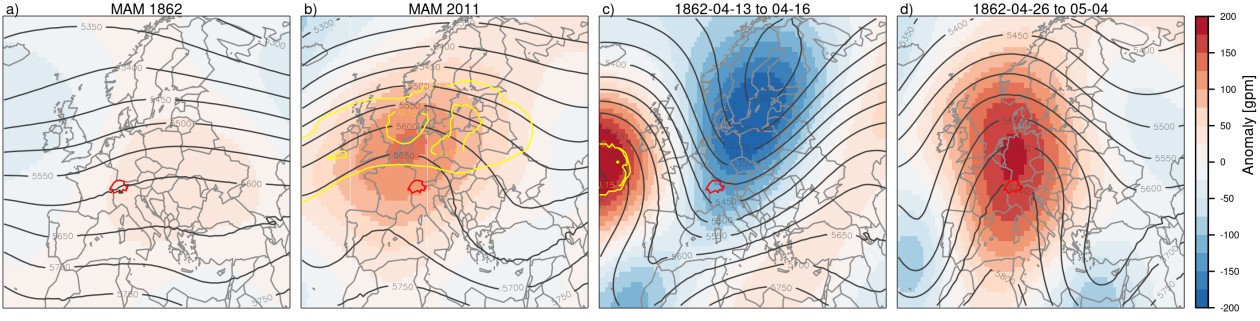

**Figure 5.** a) Geopotential height field (gpm) at the 500hPa level (contours) and its anomalies (shading) for the spring mean in 1862 and 2011. b) The period during and after a cold air outbreak in April 1862. Anomalies are calculated with respect to 1871 to 1900. Yellow lines show the blocking frequencies (0.05 and 0.1) across all time steps and all members. The data is from 20CRv3 (Slivinski et al., 2019).

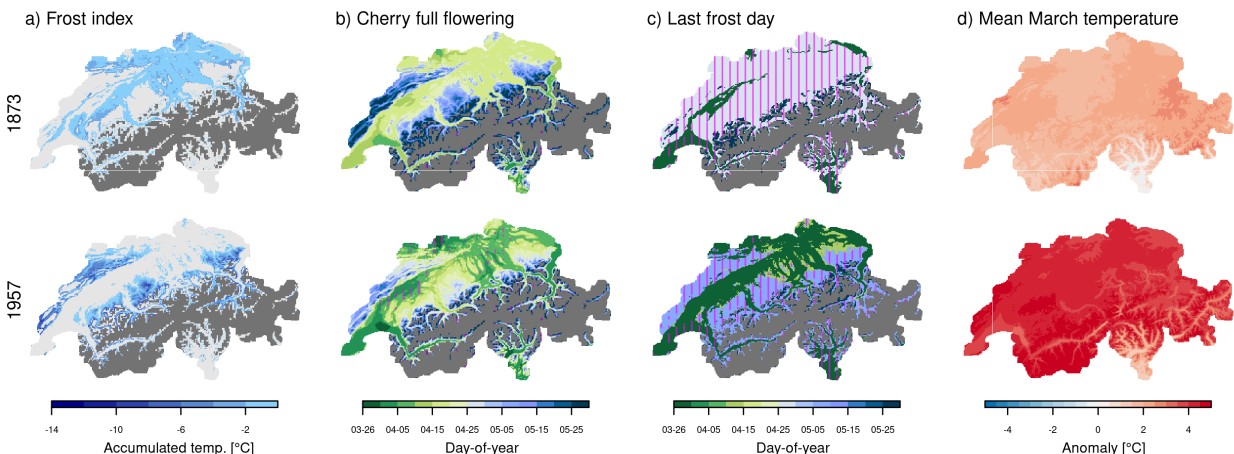

**Figure 6.** Two frost events in 1873 and 1957 causing damage to trees in Switzerland. a) frost index, b) day-of-year of cherry full flowering, c) last frost day, and d) temperature anomaly in March with respect to 1871 to 1900. The vertical purple lines in b and c) indicate areas where the last frost (cherry flowering) occurred 15 d later (earlier) than the 1871 to 1900 average. The dark grey area denotes grid cells above 1600 m a.s.l.

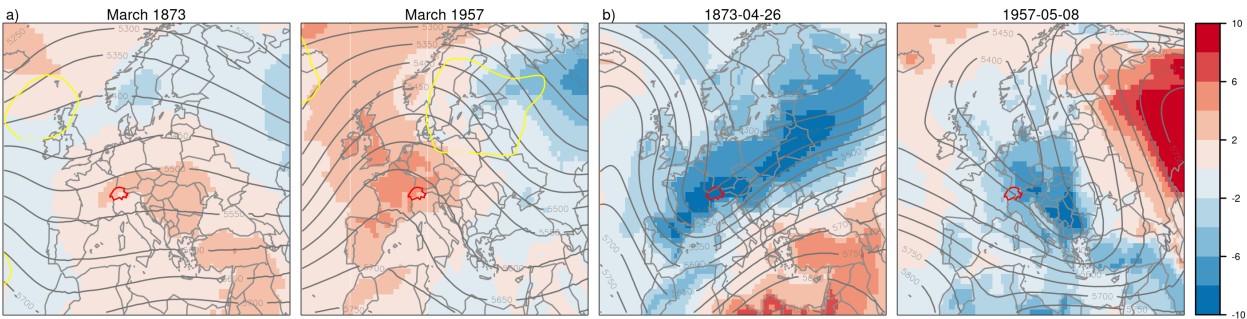

**Figure 7.** a) 2 m temperature anomalies and geopotential height fields at the 500 hPa level for March of 1873 and 1957. b) Daily 2 m temperature anomalies and daily geopotential height field of the coldest day during the late frost in Bern. Anomalies are calculated with respect to 1871 to 1900. The boundaries of Switzerland are marked in red. The data is from 20CRv3 (Slivinski et al., 2019).

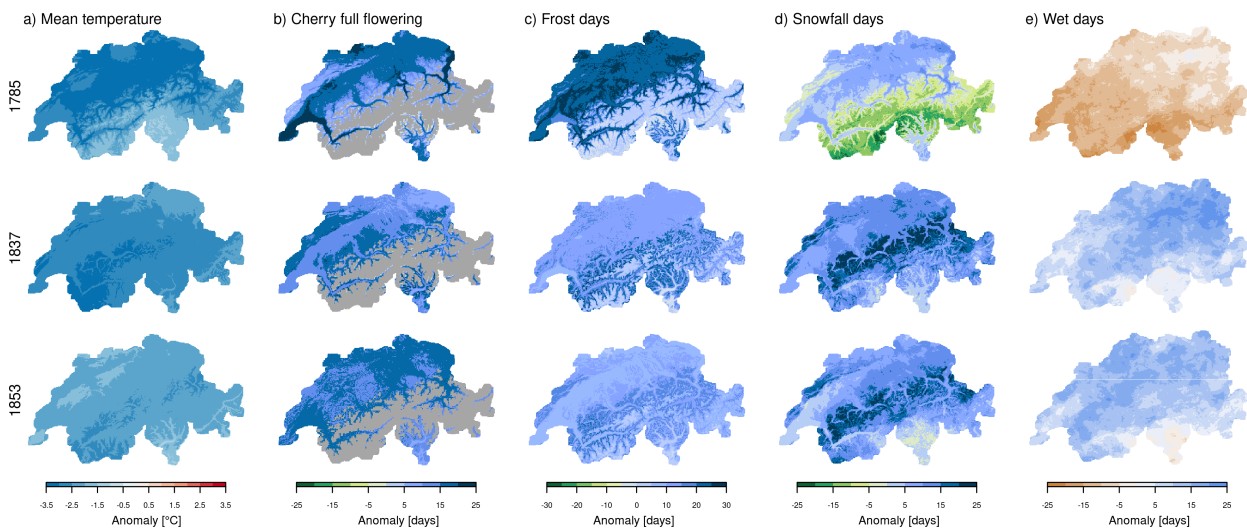

**Figure 8.** a) Spring temperature anomaly, b) anomaly in cherry flowering phenology, c) anomaly of the number of frost days, d) anomaly of the number of snow days, e) anomaly of the number of wet days for the cold springs (March to May) of 1785, 1837, 1853. All anomalies are calculated with respect to the 1871 to 1900 climatological mean. Grey areas for cherry flowering denote areas above 1600 m a.s.l.

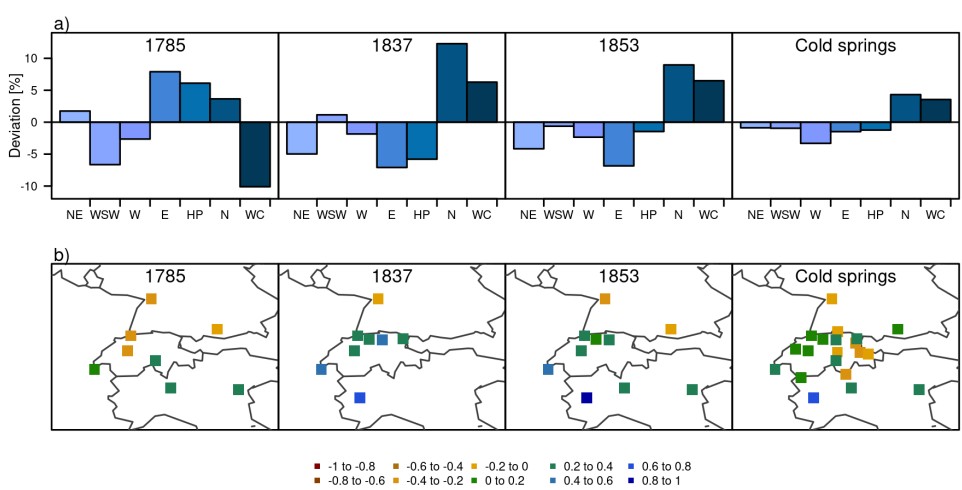

**Figure 9.** a) Anomalies of weather type frequencies for the three coldest springs and the mean of all springs below q10. The anomalies were calculated with respect to the mean frequency of weather types in March to May from 1763 to 2020 considering all weather types with a cumulative probability above 0.9. NE = Northeast, indifferent; WSW = West-southwest, cyclonic, flat pressure; W = Westerly flow over Northern Europe; E = East, indifferent; HP = High pressure over Europe; N = North, cyclonic; WC = Westerly flow over Southern Europe cyclonic (Schwander et al., 2017). b) Storminess based on pressure observations for the three coldest springs and the mean of all springs below q10. The anomaly of the standard deviation was calculated with respect to 1961 to 1990 because some series exhibit large gaps between 1871 and 1900.

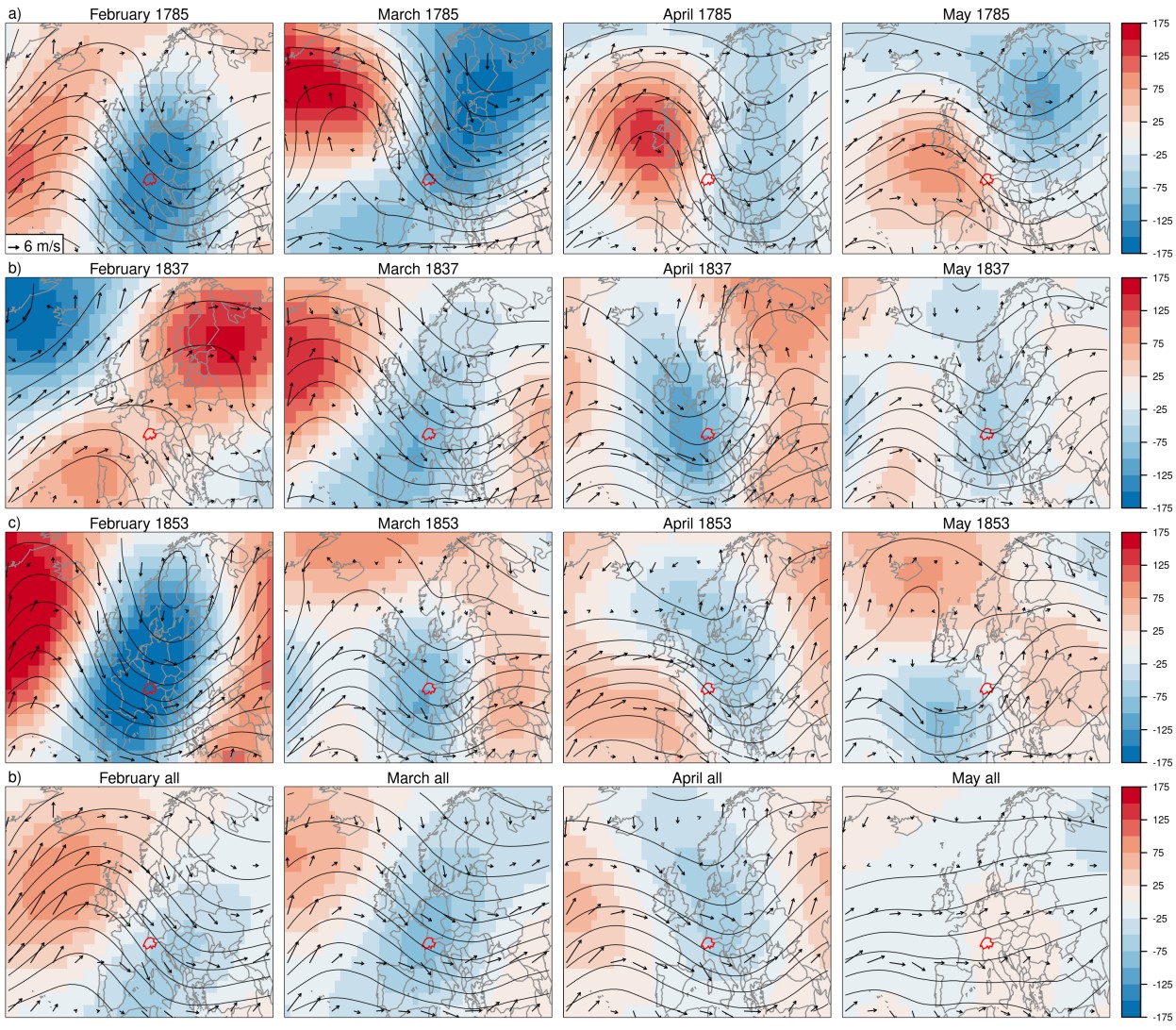

**Figure 10.** 500 hPa geopotential height field and wind direction for the months February to May of the three cold springs. Colors show anomalies with respect to the mean of 1871 to 1900, contours show absolute geopotential height field. Arrows show the wind field at 850 hPa. Switzerland is marked in darkred. a) 1785, b) 1837, c) 1853, and d) the composite of all springs below q10. The data is from ModE-RA (Valler et al., 2024).

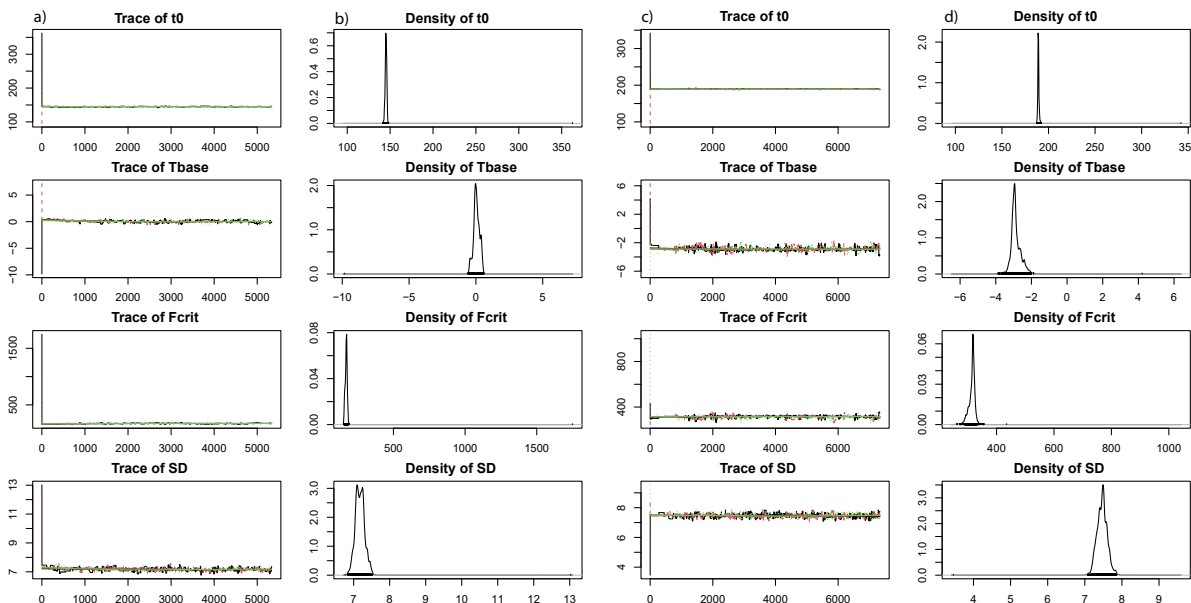

**Figure A1.** Convergence of the Markov Chain Monte Carlos DEzs algorithm for the calibration of the photo thermal time model (PTT) and the thermal time model (TT). a) Trace of the iterations after burn-in for the three chains and the four parameters. b) Marginal distribution of the four parameters based on the three chains. The first three parameters stem from the model, whereas the standard deviation is estimated during model calibration. See the formula in Eq. 2 and 1 for the parameters. Note that $t_0$ does not start on 1 January but on 21 September. c) and d) are the same as a) and b) but for the thermal time model and beech leaf unfolding. Note that we used 8000 iterations of the thermal time model to reach convergence.

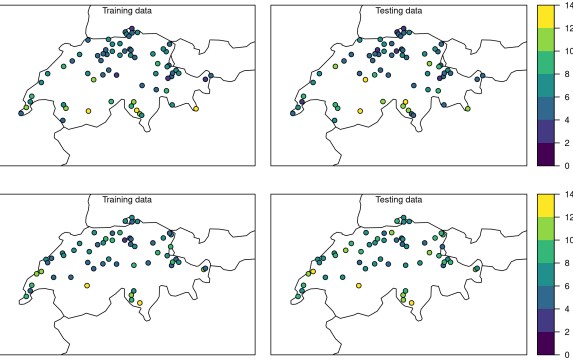

**Figure A2.** Root mean square error in days of the cross-validation for cherry flowering (upper row) and beech leaf unfolding (lower row) observations from the Swiss Phenology Network for training data (even years/left column) and testing data (odd years/right column).

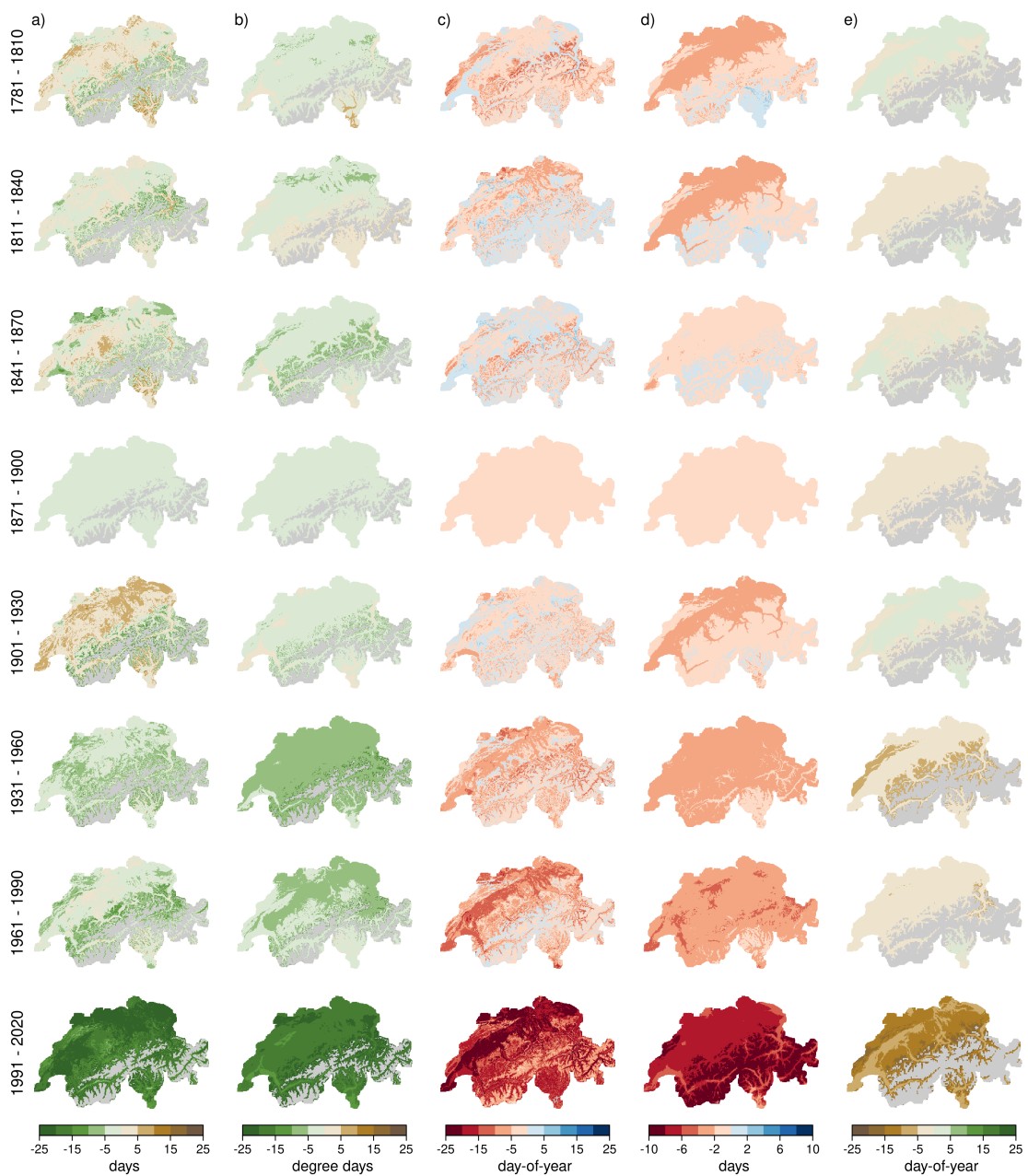

**Figure A3.** Anomalies of 30-year climatology for climate indices for all six periods with respect to the reference period from 1871 to 1900. a) Growing season start, b) growing degree days, c) last frost day, d) the number of frost days, and e) cherry full flowering. Light grey areas depict areas, where the indices are 0 in more than 75 % of the years in a period or we do not consider it because the grid cells are above 1600 m a.s.l. (last row).

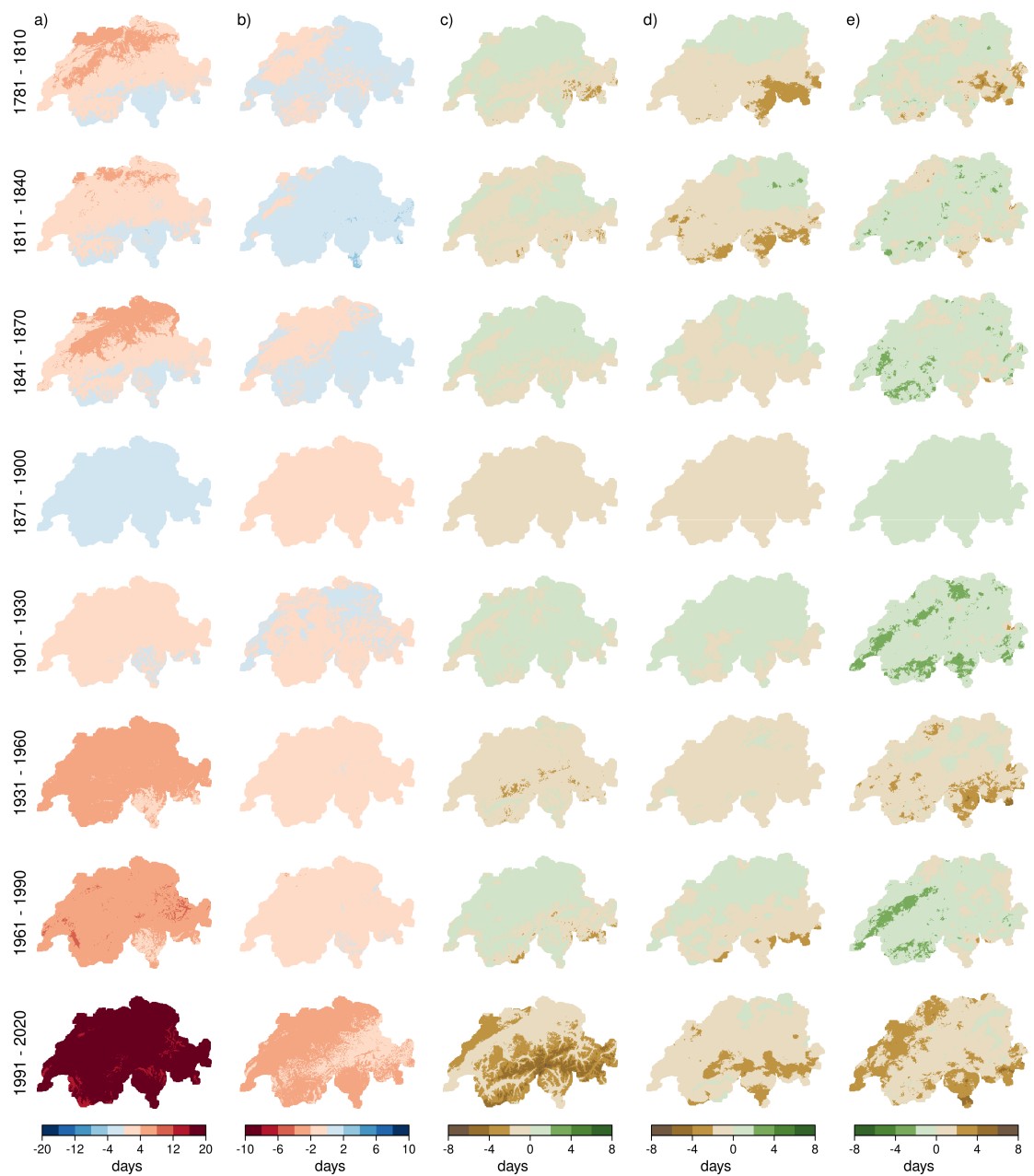

**Figure A4.** Anomalies of 30-year climatology for climate indices for all six periods with respect to the reference period from 1871 to 1900. a) Warm spell duration index, b) cold spell duration index, c) snow days, d) wet days, and e) consecutive dry days.

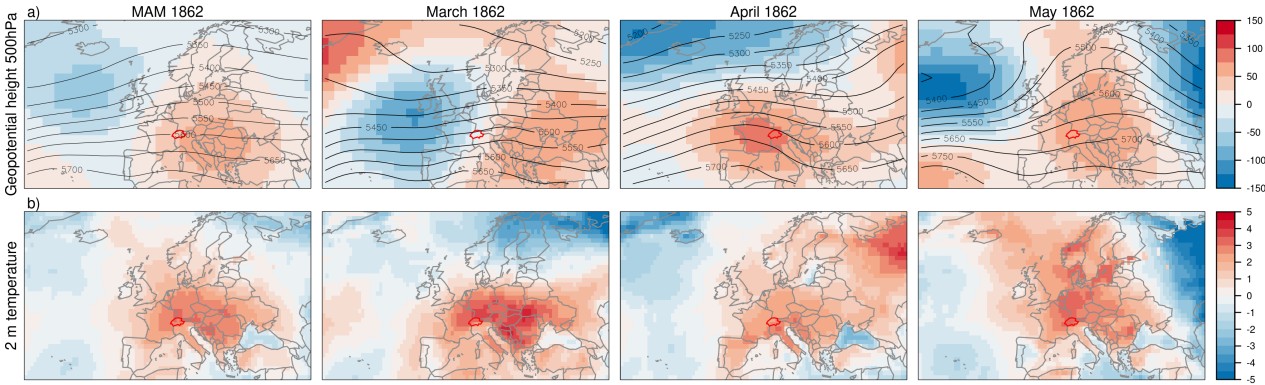

**Figure A5.** a) Geopotential height anomalies (shading) and absolute values (contour) for the 500hPa level. b) 2 m temperature anomalies. Anomalies are calculated with respect to 1871 to 1900. The borders of Switzerland are marked in darkred. Data is from ModE-RA (Valler et al., 2024).

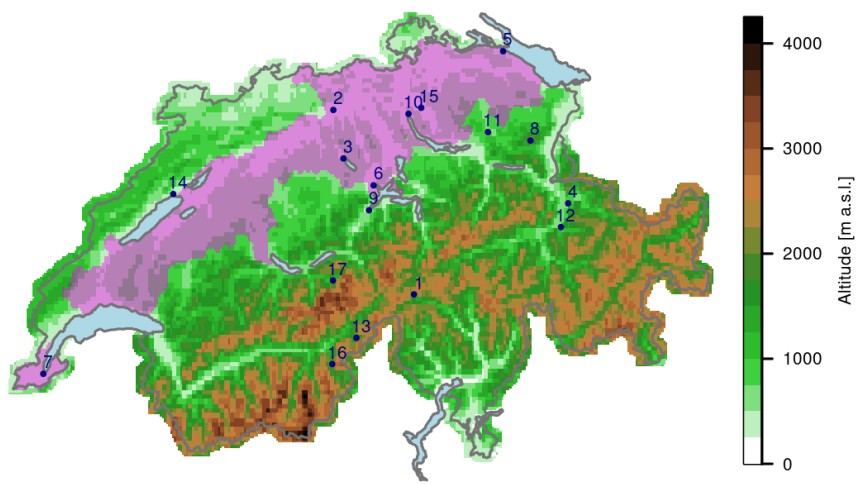

**Figure A6.** Topography and main water bodies of Switzerland. The numbers indicate weather impacts found in historical sources as listed in Table 2. The magenta shaded area indicates the area of the Swiss Plateau for which the area mean values are shown in Figure 3.

*Code availability.* The code for the calculation of indices is available on github: https://github.com/imfeldn/swiss_indices

*Data availability.* Reconstructed daily precipitation and temperature data sets over the period 1763-01-02 to 2020-12-31 are published at the open-access repository PANGAEA under https://doi.org/10.1594/PANGAEA.950236. The climate and phenological indices for the period 1763 to 2020 described in this article are published at the open-access repository PANGAEA under https://doi.org/10.1594/PANGAEA. 950236.

*Author contributions.* NI performed the analyses and wrote the manuscript. KH helped with setting up the calibration of the phenological 450 model and contributed to the manuscript. SB supervised the process and contributed to the manuscript.

*Competing interests.* The authors declare that they have no conflict of interest.

*Acknowledgements.* This work was funded by the Swiss National Science Foundation (project "WeaR", grant no. 188701) and by the European Commission through H2020 (ERC Grant PALAEO-RA 787574). The authors acknowledge the data provided by the projects "CHIMES" (SNF grant no. 169676) (Brugnara et al., 2020b), "Long Swiss Meteorological series" funded by MeteoSwiss through GCOS Switzerland 455 (Brugnara et al., 2022), and "DigiHom" (Füllemann et al., 2011).

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
