# Peer review of "Extreme springs in Switzerland since 1763 in climate and phenological indices"

_EGUsphere, 2023_

## Referee Comment (RC1)

The paper by Imfeld et al. presents several spring climatic and phenological indices for Switzerland with the aim to analyse their long-term changes since 1763 CE with particular attention to some selected warm, late frosts and cold springs. The paper has a potential to be published but the authors should consider my critical comments below.

**Major comments**

1) I understand that the paper concentrates on Switzerland, but I miss any broader Central European context, particularly in discussion of your results.

2) You are citing a lot of different places in Switzerland – it would be very useful for readers who are not so familiar with Swiss geography to see any map showing location of cited places or cantons.

3) Please consider to add some tables giving at least some characteristics (numbers) for the 30-years periods analysed which would be helpful to better follow your detail descriptions in Chapters 4.1 and 4.2.

4) Please check very carefully references – there is a lot of different errors in typos or in missing information needed for full citations.

5) Please consider the information value of Figures 1 and B1 in which you have too many very small maps.

**Minor comments**

Line 5: Please delete "for example" here.

Line 18: What is it Bise? It is not explained here or in any other parts of the manuscript, where this term is used.

Line 20: … can destroy **future** harvest …

Line 20: How spring snowfall may put trees in risk?

Line 27: In connection with late spring frost it would be useful to involve also term "false spring".

Line 28: Please add any other citations of other papers, e.g., 10.1007/s00704-023-04671-2.

Lines 29-30: Strange order of citations. Please check and present it correctly everywhere in the manuscript, where you have more than one citation (e.g., line 315).

Line 35: "dataset" – please use it uniformly as dataset (you have it also as "data set" – e.g. lines 36, 53)

Line 37: longer-term climate? Should be not long-term spring climate?

Line 58: Not clear what you mean by "wet days" here – you define them later.

Line 59: Which independent observations?

Table 1: You define snowfall days by daily mean temperature and by precipitation total, not looking on kind of precipitation? How you can be sure, that it was really snow?

Table 1: Frost index – after which phenological phase?

Lines 90-91: Is correct the term "climatological period" – it should be rather as "30-years period used as standard in climatology" or anything similar.

Line 92: I have serious doubts to speak about 1871–1900 as "pre-industrial" period. What be not better only "reference period"?

Line 106: Not clear what you mean under "quality class of at least 3".

Line 108: Where or were?

Line 115: Please use 18,000.

Figure 2: It would be useful to complement fluctuations in your characteristics by linear trends with their statistical significance.

Line 197: 10.1 °C instead of 10.09 °C.

Line 215: Fig. 4d is not there, only 4a and 4b.

Line 247: Please use Table instead of Tab.

Line 250: Please consider replacing "from March to May" by "spring", i.e. here: "Their mean spring temperature in the Swiss plateau reached …"

Figure 9 – caption: 500 hPa geopotential …

Line 310: positive **500 hPa** geopotential

Lines 371–372: Better formulation of the last sentence?

Appendix A: Variables $Rg$ and $n$ are not explained here.

Figure A1 – caption: c) and d) are the same as a) and b)

Table 2: Ct. should be explained above.

Figure C1 – caption: "absolute values (contours)" – I do not see any values there, only isolines.

Line 425: The paper was not yet published, that you are citing Climate of the Past Discussion?

Line 465: In review?

---

## Referee Comment (RC3)

[referee-annotated manuscript omitted]

---

## Author Comment (AC1)

**Response to Reviewer #1**

The paper by Imfeld et al. presents several spring climatic and phenological indices for Switzerland with the aim to analyse their long-term changes since 1763 CE with particular attention to some selected warm, late frosts and cold springs. The paper has a potential to be published but the authors should consider my critical comments below.

*We appreciate your careful review and detailed feedback. We hope that you find the following response satisfactory.*

**Major Comments**

1) I understand that the paper concentrates on Switzerland, but I miss any broader Central European context, particularly in discussion of your results.

*We will extend the discussion to a broader Central European context by adding a paragraph in the Discussion Section.*

2) You are citing a lot of different places in Switzerland – it would be very useful for readers who are not so familiar with Swiss geography to see any map showing location of cited places or cantons.

*We will add a new Figure 3 to get a better overview of where in Switzerland the described impacts happened. Also, the Figure will show the outlines of the Swiss plateau regions, for which the time series are shown in the current Figure 2.*

3) Please consider to add some tables giving at least some characteristics (numbers) for the 30-years periods analysed which would be helpful to better follow your detail descriptions in Chapters 4.1 and 4.2.

*We will add a table in the Appendix giving numbers about the 30-year periods analysed for the different indices.*

4) Please check very carefully references – there is a lot of different errors in typos or in missing information needed for full citations.

*We will correct the reference list carefully.*

5) Please consider the information value of Figures 1 and B1 in which you have too many very small maps.

*We will turn Figure 1 and B1 having the eight periods on the y axis and split the Figure according to temperature and precipitation-related indices. Thus, the maps can be increased in their size and will be better visible.*

**Minor Comments**

Line 5: Please delete "for example" here.

*We will delete it.*

Line 18: What is it Bise? It is not explained here or in any other parts of the manuscript, where this term is used.

*The Bise is a wind system occurring mainly in the Swiss plateau region between the Alps and the Jura mountains, and in some regions of eastern France. It is typically driven by a high-pressure system north or northwest of the Alps leading to cold northerly to easterly winds over the Swiss plateau. We will add a short explanation of this wind phenomenon in the manuscript based on Wanner and Furger (1990).*

Line 20: . . . can destroy future harvest . . .

*We will correct this.*

Line 20: How spring snowfall may put trees in risk?

*Due to higher temperatures, spring snow can be heavier and trees may have already developed leaves, making it easier for snow to accumulate on trees. This can increase the risk of a tree breaking a branch, for example.*

Line 27: In connection with late spring frost it would be useful to involve also term "false spring".

*Thank you for this suggestion. Since we do not further discuss "false springs", we prefer to mention only the term late spring frosts.*

Line 28: Please add any other citations of other papers, e.g., 10.1007/s00704-023-04671-2.

*Thank you. We will add more citations here.*

Lines 29-30: Strange order of citations. Please check and present it correctly everywhere in the manuscript, where you have more than one citation (e.g., line 315).

*We will correct the order of citations.*

Line 35: "dataset" – please use it uniformly as dataset (you have it also as "data set" – e.g. lines 36, 53)

*We will correct this throughout the manuscript.*

Line 37: longer-term climate? Should be not long-term spring climate?

*We will correct this.*

Line 58: Not clear what you mean by "wet days" here – you define them later.

*We will add the definition of wet days at an earlier stage in the manuscript.*

Line 59: Which independent observations?

*This sentence refers to an evaluation made in the article describing the data set (Imfeld et al., 2023). An evaluation with independent observations of precipitation occurrence showed that only up to 24% of the days were wrongly assigned to wet or dry days. Considering the daily time scale and the low availability of precipitation data, this is a good result. We will reformulate the sentence as follows to make it more clear that this refers to the article describing the data set:*

*"However, the number of monthly wet days compares well with independent observations, as shown by Imfeld et al. (2023), who compared the gridded reconstruction with an independent series from Bern."*

Table 1: You define snowfall days by daily mean temperature and by precipitation total, not looking on kind of precipitation? How you can be sure, that it was really snow?

*The two thresholds are based on the sensitivity analysis from Zubler et al. (2014) based on the 2km grid of Switzerland and they showed a good agreement between station data and grid cell snowfall days. We will recalculate the fresh snow day on our 1 km grid to evaluate the simple snow day estimate also for the higher resolved grid to determine how well this approach works. Further, we will add a sentence to stress that we look at potential fresh snow days based on these thresholds, and not at actual fresh snow days.*

Table 1: Frost index – after which phenological phase?

*We will add that we consider only the cherry full flowering.*

Lines 90-91: Is correct the term "climatological period" – it should be rather as "30-years period used as standard in climatology" or anything similar.

*We think it is adequate to talk about a climatological period in this context.*

Line 92: I have serious doubts to speak about 1871–1900 as "pre-industrial" period. What be not better only "reference period"?

*Thank you for your comment. The Swiss national weather service defined this period as their official pre-industrial reference period, also taking into account data availability and comparability to other periods and countries (Begert et al., 2019). They provide an evaluation of different reference periods and discuss the choice of 1871 to 1900, in the context of longer periods. We will state that this period is used by the Swiss national weather service as the pre-industrial reference period, but will subsequently remove the term "pre-industrial" from the manuscript.*

Line 106: Not clear what you mean under "quality class of at least 3".

*These are quality classes that have been assigned to the individual series by the study of Auchmann et al. (2018). We will add the following sentence to give a short explanation, but refer for details to the report by Auchmann et al. (2018).*

*"These quality classes were defined based on the length of records, the completeness (missing values and number of gaps longer than 5 years), reliability (number of quality flagged values), and number of inhomogeneities in the record. For further details on the quality classes see Auchmann et al. (2018)."*

Line 108: Where or were?

*We will correct this to "were".*

Line 115: Please use 18,000.

*We will correct this.*

Figure 2: It would be useful to complement fluctuations in your characteristics by linear trends with their statistical significance.

*Instead of a linear trend we will add a smoothing using a Gaussian filter to depict the decadal variability. Due to the discussed inhomogeneities in the reconstructed data set (Imfeld et al., 2023) we advise calculating linear trends only on the long observational series available e.g. Bern and Zurich (Brugnara et al., 2022).*

Line 197: 10.1 °C instead of 10.09 °C.

*We will correct this.*

Line 215: Fig. 4d is not there, only 4a and 4b.

*We will correct this in the text.*

Line 247: Please use Table instead of Tab.

*We will change Tab. to Table.*

Line 250: Please consider replacing "from March to May" by "spring", i.e. here: "Their mean spring temperature in the Swiss plateau reached ..."

*We will correct this occurrence, and also change "March to May" to "spring" in other occurrences where it is more appropriate.*

Figure 9 – caption: 500 hPa geopotential ...

*We will correct the caption.*

Line 310: positive 500 hPa geopotential

*We will correct this sentence.*

Lines 371–372: Better formulation of the last sentence?

*We will reformulate this sentence.*

Appendix A: Variables Rg and n are not explained here.

*We will add an explanation of these two variables and move the Appendix section to the main manuscript (phenological application) to expand the description of the phenological models as suggested by another reviewer.*

Figure A1 – caption: c) and d) are the same as a) and b)

*We will correct this caption.*

Table 2: Ct. should be explained above.

*We will add an explanation of Ct. (Canton) in the Table caption.*

Figure C1 – caption: "absolute values (contours)" – I do not see any values there, only isolines.

*The isolines represent the absolute values of the geopotential height field at 500 hPa. Indeed, the values of the isolines are not visible. We adjusted this in the revised figure (see below).*

Line 425: The paper was not yet published, that you are citing Climate of the Past Discussion?

*We will correct this reference.*

Line 465: In review?

*This is the data set corresponding to the herein presented manuscript. It is possible to evaluate it as well and it will be published alongside the manuscript. Currently, the DOI is not available. We will remove the "in review" and provide the correct DOI as soon as it is available.*

[Figure]

Figure 1: a) Geopotential height (m) anomalies (shading) and absolute values (contour) at the 500hPa level. b) 2 m temperature anomalies. Anomalies are calculated with respect to 1871 to 1900. The borders of Switzerland are marked in darkred. Data is from ModE-RA (Valler et al., in review).

**References**

Auchmann, R., Brugnara, Y., Rutishauser, T., Brönnimann, S., Gehrig, R., Pietragalla, B., Begert, M., Sigg, C., Knechtl, V., Calpini, B., and Konzelmann, T.: Quality Analysis and Classification of Data Series from the Swiss Phenology Network, Technical Report Meteoswiss, 271, 77 pp, URL https://www.meteoschweiz.admin.ch/dam/jcr:5220f36c-435d-4d46-b766-ba91b0dffe37/fachbericht-271.pdf, (last accessed 20 June 2023), 2018.

Begert, M., Stöckli, R., and Croci-Maspoli, M.: Klimaentwicklung in der Schweiz - vorindustrielle Referenzperiode und Veränderung seit 1864 auf Basis der Temperaturmessung, Technical Report Meteoswiss, 274, 23 pp, URL https://www.meteoswiss.admin.ch/dam/jcr:4c89a839-d577-47f1-aeb9-a30749ddaf2b/AB_Vorind_Refp_v1.1_de.pdf, (last accessed 30 June 2023), 2019.

Brugnara, Y., Hari, C., Pfister, L., Valler, V., and Brönnimann, S.: Pre-industrial temperature variability on the Swiss Plateau derived from the instrumental daily series of Bern and Zurich, Climate of the Past, 18, 2357–2379, https://doi.org/10.5194/cp-18-2357-2022, 2022.

Imfeld, N., Pfister, L., Brugnara, Y., and Brönnimann, S.: A 258-year-long data set of temperature and precipitation fields for Switzerland since 1763, Climate of the Past, 19, 703–729, https://doi.org/10.5194/cp-19-703-2023, 2023.

Valler, V., Franke, J., Brugnara, Y., Samakinwa, E., Hand, R., Lundstad, E., Burgdorf, A.-M., and Brönnimann, S.: ModE-RA – A global monthly paleo-reanalysis of the modern era (1421 to 2008), Sci. Data, in review.

Wanner, H. and Furger, M.: The bise—climatology of a regional wind north of the Alps, Meteorology and Atmospheric Physics, 43, 105–115, 1990.

Zubler, E. M., Scherrer, S. C., Croci-Maspoli, M., Liniger, M. A., and Appenzeller, C.: Key climate indices in Switzerland; expected changes in a future climate, Climatic change, 123, 255–271, https://doi.org/10.1007/s10584-013-1041-8, 2014.

---

## Author Comment (AC2)

**Response to Reviewer #3**

This is overall well-written, and the approaches are appropriate, I have provided an annotated copy of the manuscript to assist the authors in making corrections. I have three substantive points below which require addressing.

*Thank you for your detailed review of our manuscript and your constructive comments. In the following, we address the three substantive points raised by you. We are also thankful for the remarks made directly in the manuscript.*

1) The snow days approach presented is not actually documenting snow days – you are presenting 'potential snow days'. What you are presenting is a 'potential snowfall days', as no observation of snowfall are made. This may seem rather pedantic but is important from an accuracy perspective, and the method you apply is likely to result in substantial overestimation of snow days. I recommend rephrasing the snow days to potential or removing this from the paper as it is likely to result in an ill-informed discussion. The sensitivity of such an approach in a country with extensive elevation differences is also liable to result in high uncertainties. I would remove this section from the paper as it I fear detracts from the paper overall, an alternative approach would be to demonstrate that the method you apply provides a good proxy for snow days for the different regions using observational records.

*Thank you for this comment. We use a very simple approach to deduce days with snowfall from daily precipitation and daily mean temperature. Such a simple approach is needed because we only have these two variables available for the period back to 1763. The evaluation of Zubler et al. (2014) for the 2x2 km grid and 46 stations of the National Basic Climatological Network shows high correlations of 0.93 and a mean bias of -0.23 days stating, however, that the biases may come from the smoothed 2-km topography. We suggest recalculating this evaluation for our 1x1 km grid to evaluate the bias on the higher resolved grid. However, we will keep the snowfall days index in the manuscript because we think that an estimation of snowfall days in spring is relevant for illustrating changes in spring weather in Switzerland.*

*Therefore, we will add the new results of the snowfall evaluation in a short paragraph and we will emphasise that what we are showing are not actual snowfall days, but days when snowfall could potentially occur.*

2) I am concerned by your use of the period 1871-1900 as pre-industrial, I appreciate that parts of Switzerland may not have been industrial, but certainly neighbouring regions and countries had extensive industry by this stage. I think using any period post-1750/1800 as pre-industrial is fraught with risk, particularly when dealing with climate and climatic parameters which are influenced beyond regional boundaries.

*This period has been suggested by the Swiss national weather service based on evaluations using different periods and taking into consideration comparability to other periods and countries (Begert et al., 2019). Thus, we used the same period for consistency with respect to other publications by the Swiss national weather service. We suggest still using the 1871 to 1900 period but referring to it subsequently in the manuscript as a reference period, and not a pre-industrial reference period.*

3) The discussion would benefit from a section explaining how the results of this paper relate to studies in neighbouring regions/countries, can you provide a little further context please of how this fits with other studies in Central Europe.

*We will add a paragraph on how our study relates to studies in Central Europe. However, please note that only a few studies are available for the 18th and 19th centuries, especially focusing on spring.*

*Furher, we will correct the spelling mistakes marked directly in the manuscript. Also, we decided to change Figure 1 by creating two Figures covering the different indices. The anomalies with respect to the period 1871 to 1900 are already provided in the Appendix. We do not evaluate whether statistical step changes were present around the 1980s, since several studies have been published (e.g. Marty, 2008; Reid et al., 2016) and possible explanations provided (Sippel et al., 2020).*

*Thank you also for the suggestion of further literature. We will include these.*

**References**

Begert, M., Stöckli, R., and Croci-Maspoli, M.: Klimaentwicklung in der Schweiz - vorindustrielle Referenzperiode und Veränderung seit 1864 auf Basis der Temperaturmessung, Technical Report Meteoswiss, 274,

23 pp, URL `https://www.meteoswiss.admin.ch/dam/jcr:4c89a839-d577-47f1-aeb9-a30749ddaf2b/AB_Vorind_Refp_v1.1_de.pdf`, (last accessed 30 June 2023), 2019.

Marty, C.: Regime shift of snow days in Switzerland, Geophysical Research Letters, 35, https://doi.org/10.1029/2008GL033998, 2008.

Reid, P. C., Hari, R. E., Beaugrand, G., Livingstone, D. M., Marty, C., Straile, D., Barichivich, J., Goberville, E., Adrian, R., Aono, Y., Brown, R., Foster, J., Groisman, P., Hélaouët, P., Hsu, H.-H., Kirby, R., Knight, J., Kraberg, A., Li, J., Lo, T.-T., Myneni, R. B., North, R. P., Pounds, J. A., Sparks, T., Stübi, R., Tian, Y., Wiltshire, K. H., Xiao, D., and Zhu, Z.: Global impacts of the 1980s regime shift, Global Change Biology, 22, 682–703, https://doi.org/10.1111/gcb.13106, 2016.

Sippel, S., Fischer, E. M., Scherrer, S. C., Meinshausen, N., and Knutti, R.: Late 1980s abrupt cold season temperature change in Europe consistent with circulation variability and long-term warming, Environmental Research Letters, 15, 094 056, https://doi.org/10.1088/1748-9326/ab86f2, 2020.

Zubler, E. M., Scherrer, S. C., Croci-Maspoli, M., Liniger, M. A., and Appenzeller, C.: Key climate indices in Switzerland; expected changes in a future climate, Climatic change, 123, 255–271, https://doi.org/10.1007/s10584-013-1041-8, 2014.

---

## Author Response (AR1)

**Response to Reviewer #1**

The paper by Imfeld et al. presents several spring climatic and phenological indices for Switzerland with the aim to analyse their long-term changes since 1763 CE with particular attention to some selected warm, late frosts and cold springs. The paper has a potential to be published but the authors should consider my critical comments below.

*We appreciate your careful review and detailed feedback. We hope that you find the following response satisfactory.*

**Major Comments**

1) I understand that the paper concentrates on Switzerland, but I miss any broader Central European context, particularly in discussion of your results.

*We extended the discussion by adding three small paragraphs in the discussion section, considering studies for Italy and the Czech Republic, and for a general Central Europe context.*

2) You are citing a lot of different places in Switzerland – it would be very useful for readers who are not so familiar with Swiss geography to see any map showing location of cited places or cantons.

*Because there are already a lot of figures in the manuscript, we added the new figure to the Appendix, now Figure A6. It shows the locations of the reported impacts, the topography of Switzerland, and the outline of the Swiss Plateau area. The numbers are marked in Table 2. Also, we included the historical sources for the warm spring of 1862 in Table 2.*

3) Please consider to add some tables giving at least some characteristics (numbers) for the 30-years periods analysed which would be helpful to better follow your detail descriptions in Chapters 4.1 and 4.2.

*We decided not to add such a table to the manuscript, but instead increased the Figures of the 30-year climatological maps and also made the time series of the area mean of the Swiss Plateau better visible. We hope that this also helps to better follow the descriptions in Chapter 4. We think that an additional table in the manuscript might further complicate the discussion.*

4) Please check very carefully references – there is a lot of different errors in typos or in missing information needed for full citations.

*We checked the references and corrected the errors we found.*

5) Please consider the information value of Figures 1 and B1 in which you have too many very small maps.

*We turned Figure 1 and B1 having the eight periods on the y axis and split the Figures into two Figures (new Figure 1 and 2, and A3 and A4). Thus, the individual maps can be increased in their size and are better visible.*

**Minor Comments**

Line 5: Please delete "for example" here.

*We deleted it.*

Line 18: What is it Bise? It is not explained here or in any other parts of the manuscript, where this term is used.

*We added the following paragraph to the main manuscript:*

*"The Bise is a wind in the Alpine area channeled between the Jura mountains and the Alps, and is related to high pressure and anticyclonic weather conditions (Wanner and Furger, 1990). It is further associated with the advection of cold and dry continental air, and the co-occurrence of stratus formation."*

Line 20: . . . can destroy future harvest . . .

*We corrected this.*

Line 20: How spring snowfall may put trees in risk?

*We added two citations that refer to how snowfall can put trees at risk.*

Line 27: In connection with late spring frost it would be useful to involve also term "false spring".

*Thank you for this suggestion. Since we do not further discuss the term "false springs", we did not add further context about this term, but we mentioned it in relation to the article you suggested in the following comment.*

Line 28: Please add any other citations of other papers, e.g., 10.1007/s00704-023-04671-2.

*Thank you. We added the citation you mentioned on past and present risk of spring frost.*

Lines 29-30: Strange order of citations. Please check and present it correctly everywhere in the manuscript, where you have more than one citation (e.g., line 315).

*We corrected the order of citations in the entire manuscript.*

Line 35: "dataset" – please use it uniformly as dataset (you have it also as "data set" – e.g. lines 36, 53)

*We corrected this throughout the manuscript.*

Line 37: longer-term climate? Should be not long-term spring climate?

*We changed this to long-term spring climate.*

Line 58: Not clear what you mean by "wet days" here – you define them later.

*We added the definition of wet days, when the term first appears in the manuscript.*

Line 59: Which independent observations?

*This sentence refers to an evaluation made in the article describing the data set (Imfeld et al., 2023). An evaluation with independent observations of precipitation occurrence showed that only up to 24% of the days were wrongly assigned to wet or dry days. Considering the daily time scale and the low availability of precipitation data, this is a good result. We reformulated the sentence as follows to make it more clear that this refers to the article describing the data set:*

*"However, the number of monthly wet days compares well with independent observations, as shown by Imfeld et al. (2023), who compared the gridded reconstruction with an independent series from Bern."*

Table 1: You define snowfall days by daily mean temperature and by precipitation total, not looking on kind of precipitation? How you can be sure, that it was really snow?

*The two thresholds are based on the sensitivity analysis from Zubler et al. (2014) based on the 2km grid of Switzerland and they showed a good agreement between station data and grid cell snowfall days based on inverse distance weighting. We compared the number of snowfalls days estimated from our 1 km grid to 27 observational series of snowfall days across Switzerland for the spring season. With a mean Pearson correlation of 0.81 and a mean bias of -0.28, the results were slightly worse than in Zubler et al. (2014). However, Zubler et al. (2014) evaluated snowfall days for an entire year, whereas we considered the spring season only. Estimating snowfall days in spring based on such threshold may be more difficult than in e.g. winter which would explain the difference in results. Also, we think it would be valuable, to more in-depth evaluate the snowfall estimates, however, this is beyond the scope of this article. We added the following short paragraph to the manuscript describing the evaluation results.*

*"We evaluated these thresholds by comparing the closest grid cells of snowfall days with observations of snowfall days at 27 different stations across Switzerland for the spring season. The results showed a mean Spearman correlation of 0.8 and a mean bias of -0.3 d. Especially for stations above 1000 m, the gridded data set showed a tendency to underestimate the number of snowfall days. It is therefore important to remember that our snowfall days only represent potential snowfall days. Nevertheless, these estimates provide a good basis for evaluating spring snowfall over the historical period."*

Table 1: Frost index – after which phenological phase?

*We added that we considered only the cherry full flowering.*

Lines 90-91: Is correct the term "climatological period" – it should be rather as "30-years period used as standard in climatology" or anything similar.

*We think it is adequate to talk about a climatological period in this context.*

Line 92: I have serious doubts to speak about 1871–1900 as "pre-industrial" period. What be not better only "reference period"?

*This period has been suggested by the Swiss national weather service based on evaluations using different periods and taking into consideration comparability to other periods and countries (Begert et al., 2019). Thus, we used the same period for consistency with respect to other publications by the Swiss national weather service. We kept the 1871 to 1900 period as a reference period mentioning that it is used by the national weather service as a pre-industrial reference, but we removed the term pre-industrial from the rest of the manuscript.*

Line 106: Not clear what you mean under "quality class of at least 3".

*These are quality classes that have been assigned to the individual series by the study of Auchmann et al. (2018). We added the following sentence to give a short explanation, but refer for details to the report by Auchmann et al. (2018).*

*"These quality classes were defined based on the length of records, the completeness (missing values and number of gaps longer than 5 years), reliability (number of quality flagged values), and number of inhomogeneities in the record. For further details on the quality classes see Auchmann et al. (2018)."*

Line 108: Where or were?

*We corrected this.*

Line 115: Please use 18,000.

*We corrected this.*

Figure 2: It would be useful to complement fluctuations in your characteristics by linear trends with their statistical significance.

*Instead of a linear trend we added a smoothing using a Gaussian filter to depict the decadal variability. Due to the discussed inhomogeneities in the reconstructed data set (Imfeld et al., 2023) we prefer calculating linear trends only on the long observational series available e.g. Bern and Zurich (Brugnara et al., 2022).*

Line 197: 10.1 °C instead of 10.09 °C.

*We corrected this.*

Line 215: Fig. 4d is not there, only 4a and 4b.

*We corrected the figure labelling.*

Line 247: Please use Table instead of Tab.

*We changed this.*

Line 250: Please consider replacing "from March to May" by "spring", i.e. here: "Their mean spring temperature in the Swiss plateau reached . . . "

*We corrected this occurrence, and also changed "March to May" to "spring" in other occurrences where it seemed more appropriate.*

Figure 9 – caption: 500 hPa geopotential . . .

*We corrected the caption.*

Line 310: positive 500 hPa geopotential

*We corrected this sentence.*

Lines 371–372: Better formulation of the last sentence?

*We deleted the last sentence and reformulated the second last sentence.*

Appendix A: Variables Rg and n are not explained here.

*We will add an explanation of these two variables and move the Appendix section to the main manuscript (phenological application) to expand the description of the phenological models as suggested by another reviewer.*

Figure A1 – caption: c) and d) are the same as a) and b)

*We corrected the caption.*

Table 2: Ct. should be explained above.

*We added an explanation of Ct. (Canton) in the Table caption.*

Figure C1 – caption: "absolute values (contours)" – I do not see any values there, only isolines.

*The isolines represent the absolute values of the geopotential height field at 500 hPa. Indeed, the values of the isolines were not visible. We adjusted this in the revised figure (see below).*

[Figure]

Figure 1: a) Geopotential height (m) anomalies (shading) and absolute values (contour) at the 500hPa level. b) 2 m temperature anomalies. Anomalies are calculated with respect to 1871 to 1900. The borders of Switzerland are marked in darkred. Data is from ModE-RA (Valler et al., in review).

Line 425: The paper was not yet published, that you are citing Climate of the Past Discussion?

*We corrected this reference.*

Line 465: In review?

*This is the data set corresponding to the herein presented manuscript. It is possible to evaluate it as well and it will be published alongside the manuscript. Currently, the DOI is not yet available. We will remove the "in review" and provide the correct DOI as soon as it is available.*

**References**

Auchmann, R., Brugnara, Y., Rutishauser, T., Brönnimann, S., Gehrig, R., Pietragalla, B., Begert, M., Sigg, C., Knechtl, V., Calpini, B., and Konzelmann, T.: Quality Analysis and Classification of Data Series from the Swiss Phenology Network, Technical Report Meteoswiss, 271, 77 pp, URL `https://www.meteoschweiz.admin.ch/dam/jcr:5220f36c-435d-4d46-b766-ba91b0dffe37/fachbericht-271.pdf`, (last accessed 20 June 2023), 2018.

Begert, M., Stöckli, R., and Croci-Maspoli, M.: Klimaentwicklung in der Schweiz - vorindustrielle Referenzperiode und Veränderung seit 1864 auf Basis der Temperaturmessung, Technical Report Meteoswiss, 274, 23 pp, URL `https://www.meteoswiss.admin.ch/dam/jcr:4c89a839-d577-47f1-aeb9-a30749ddaf2b/AB_Vorind_Refp_v1.1_de.pdf`, (last accessed 30 June 2023), 2019.

Brugnara, Y., Hari, C., Pfister, L., Valler, V., and Brönnimann, S.: Pre-industrial temperature variability on the Swiss Plateau derived from the instrumental daily series of Bern and Zurich, Climate of the Past, 18, 2357–2379, https://doi.org/10.5194/cp-18-2357-2022, 2022.

Imfeld, N., Pfister, L., Brugnara, Y., and Brönnimann, S.: A 258-year-long data set of temperature and precipitation fields for Switzerland since 1763, Climate of the Past, 19, 703–729, https://doi.org/10.5194/cp-19-703-2023, 2023.

Valler, V., Franke, J., Brugnara, Y., Samakinwa, E., Hand, R., Lundstad, E., Burgdorf, A.-M., and Brönnimann, S.: ModE-RA – A global monthly paleo-reanalysis of the modern era (1421 to 2008), Sci. Data, in review.

Wanner, H. and Furger, M.: The bise—climatology of a regional wind north of the Alps, Meteorology and Atmospheric Physics, 43, 105–115, 1990.

Zubler, E. M., Scherrer, S. C., Croci-Maspoli, M., Liniger, M. A., and Appenzeller, C.: Key climate indices in Switzerland; expected changes in a future climate, Climatic change, 123, 255–271, https://doi.org/10.1007/s10584-013-1041-8, 2014.

**Response to Reviewer #2**

Review of 'Extreme springs in Swizerland since 1763 in climate and phenological indices' by Noemi Imfeld, Koen Hufkens and Stefan Bronnimann

Based on an existing reconstruction of daily mean temperature and daily precipitation amounts, an overview is given of the variations in sping climate over the period since 1763 for Switzerland. The analysis is based on climatological indices and two phenological indices. In the study, the impact of climate change on these indices is documented and the climatic variability in earlier times is highlighted. A few exceptional springs are discussed in detail and in relation to the atmospheric situation.

The manuscript is well written and a joy to read. There are - as far as I can say - no methodological errors. The main concerns relate to the presentation of results which could be a bit more clear, and the selection and presentation of the climate and phenological indices needs some further thoughts. In addition, the assessment of uncertainty in the results deserves more attention.

My advise to the editor is to accept with minor revisions.

*Thank you for your detailed and constructive review of our manuscript. We hope that you find the following response satisfactory.*

**Main Comments**

*) In contrast to the warming spring, the warming of the winter climate has a delaying effect on spring phenology and Wang et al. (2020) argue that existing winter chilling model underestimate the effects of winter chilling, leading to substantial overestimates of the advance of spring phenology under climate change. A similar concern relates to the cherry flowering model and the beech leaf unfolding model. The model used to relate the cherry flowering and beech leaf unfolding is not completely clear to me, but it seems that winter chilling is not part of the equation. Motivate why the winter chilling is left out or explicitly comment on this aspect - if possible with an assessment of the consequence of not using winter chilling.

The description of the phenological model is very terse. As this is an important - and interesting! - part of the work, the description of the model should be expanded a bit to guide the readers through the model that are less well acquainted with these models.

*Thank you for this relevant comment. We evaluated different phenological models for the two phenological phases which are all based on daily mean temperatures and which are implemented in the phenor R package by Hufkens et al. (2018). This also includes models with winter chilling (the alternating model and chilling degree days model, see the models in Hufkens et al., 2018). Most models showed largely similar behaviours. We agree, however, that not considering winter chilling may affect the model estimates of phenological dates. Our models are calibrated based on data from 1950 to 2020, however with more observations in the later years. The last around 30 years showed a relevant advancement of spring phenology especially for cherry flowering with likely lower chilling accumulations in winter. Calibrating a model on such data without considering winter chilling might lead to a higher heat requirement as shown in Wang et al. (2020). Further, transferring these models to the past could lead to phenological dates being later in spring. This is hypothetical, and a comprehensive study on such effects would be needed. In this article, we focus, however, on providing a first estimate of phenological phases based on a commonly used processed-based model for the past 258 years. Providing a comprehensive study of phenological models and effects of different models for the past is highly relevant, but out of scope for this article.*

*To make the section about the phenological application more clear, we expanded the description of the phenological model by including the current Appendix A in Section 3.2. Further, we discussed the individual terms in the equations of the phenological models in more detail, and also briefly discussed the effect of model selection (e.g. lack of winter chilling) and model calibration period for estimating phenological dates of the past few hundred years.*

*) the selection of climatic indices is strongly biased towards the temperature-related indices. The only two indices which are precipitation-based are the number of Wet Days and Snowfall Days (the latter is a mix between precip and temperature). Although these two indices are relevent, it would be interesting to add indices that relate to droughts or pluvials - like the Consecutive Dry Days or Consecutive Wet Days indices. This would contribute to earlier studies on droughts where a propagating signal from spring drought into summer drought

is observed, and might give some perspective on e.g. the drought in the mid 1940s in central Europe (Brazdil et al. 2016; Hirschi et al. 2013)

*We primarily focused on the temperature indices because the precipitation reconstruction shows a much lower reliability than the temperature reconstruction (see Imfeld et al., 2023) and we do not want to promote using the precipitation reconstruction in an imprudent way. We split Figure 1 into two Figures which allowed us to increase the size of the individual maps, and we added the Consecutive Dry Days index since it is also based on wet days and a relevant index for spring weather.*

*) In the discussion of the quality of the reconstruction, it was noted that the skill in the temperature reconstruction is higher than that of precipitation. This is perfectly understandable, but what is missing is a view how this uncertainty propagates into the indices. It would have been very nice if the authors would be able to assess the uncertainty in the indices, and therefore in the conclusions. I briefly went through the paper that documents the reconstruction, but I understand that this rerconstruction does not come with an uncertainty estimate in terms of an ensemble? That would have made the assessment of uncertainty not too difficult (it only requires quite a bit of computations). The uncertainty assessment in the manuscript is now based on using various sources reconstructions (like 20CR, ModE-RA and long observational records). The spread in the various reconstructions is demonstrated in fig. 2, but the sometimes large deviations between the estimates is not discussed. Particularly the cherry flowering in the Liestal deserves some attentiuon as the observations show much stronger variability and show for many years much earlier flowering. Can you indicate if this discrepancy relates to the temperature reconstruction or is there an issue with the phenological model?

*Thank you for this comment. This comment addresses two different points, a) the uncertainty in the original reconstruction and b) the deviations in the cherry flowering estimates concerning an observational series. We answer these two points individually.*

*a) In the article about the data set (Imfeld et al., 2023), we described in detail how the reconstruction was performed. This includes an ensemble based on the 50 best analogue days which was then used in the Ensemble Kalman Fitting for the temperature reconstruction (for details, please refer to the article). However, this is not a true ensemble but an ordered ensemble, where the first analogue day (based on its Gower distance) is a better representation of the historical period than e.g. the second analogue day. A true ensemble would be needed to do a proper uncertainty analysis.*

*To provide an idea of the uncertainty in the reconstruction, we added to the phenological estimate of Liestal in Figure 3, the ensemble of the 10 best analogue days (incl. temperature assimilation) while emphasising that this is not equivalent to a true ensemble. As stated in the updated manuscript, the individual ensemble members only vary between -5 and +4 days in their phenological estimate.*

*b) Comparing the reconstructed cherry flowering with the cherry flowering series of Liestal showed a mean bias of 7.36 days as stated in the manuscript. The phenological estimates were based on a phenological model calibrated using cherry flowering observations across Switzerland and then applied to each grid cell of the temperature reconstruction. A direct comparison of the closest grid cell from our reconstructed cherry flowering and the cherry flowering series of Liestal can be biased for several reasons. For example, the exposition and micro-climate of the tree (within the 1 km grid cell) can be different leading to e.g. warmer temperatures at the location of the tree and leading to earlier flowering. Also, tree-specific characteristics can lead to a flowering different from when a model is calibrated using all cherry flowering trees in Switzerland likewise leading to biases for this specific tree.*

Interesting is also that the GGD and mean temperature reconstructions are spot-on with the 20CR after ±1840, but for the earliest decades, there is a bit of a bias. Can you comment on this?

*We assume that this difference in the early years mainly stems from the 20CRv3 reanalysis. The first years of the 20CRv3 reanalysis from 1806 to 1835 are an experimental extension (Slivinski et al., 2021). In our case the results seem to deviate more strongly from observations for the experimental extension of the reanalysis.*

**Other concerns the authors may want to look at**

*) line 95: The group involved in the ETCCDI also prescribed levels of missing data that are allowed in the aggregation of seasonal/annual values. Can you argue why you deviate from their approach by selecting the 10% threshold?

*We used the 10% threshold because gaps often occur for several consecutive days, which could affect the seasonal aggregated value by e.g. missing a specific synoptic situation. Note, that this threshold only applies to the Swiss*

*plateau series (Brugnara et al., 2022) and thus, is only relevant for a comparative purpose in Figure 2. The Swiss reconstruction is a spatially and temporally complete data set.*

\*) Figure 1: Except for the warm spell duration index and cold spell duration index (figs. e and f), the climatologies of the 30-yr periods are quite similar. This has been noted in the text. The figure would be a bit more interesting if you would show one reference period (e.g. your favourite 1871-1900 period) and deviations from this reference for the other periods.

*One of the main messages of this Figure is that the 30-year periods are quite similar to each other, except for the last few decades. To show the differences in more detail we added in the Appendix a Figure showing the difference between all periods and the 1871-1900 period.*

\*) figure 5: it seems that in the figures for the Last Frost Day, something is wrong. I see purple vertical stripes and I wonder if some detail is lacking in figure a?

*Nothing is missing in the Figure. Please, see the caption: The vertical purple lines indicate areas where frost (cherry flowering) occurred 15 d later (earlier) than the 1871 to 1900 average.*

**Smallish concerns**

\*) line 81: for completeness, you could mention that your definition is considered colder than what is usually called a frost, but also warmer than an ice day (where the Tmax drops below zero)

*Thank you, we added this in a sentence.*

\*) line 197: the use of two digits for temperature is not in-line with what is used in the rest of the manuscript.

*We corrected this.*

\*) figure 5: the labels of the colour bars are difficult to read with this size of the figure (I needed to zoom to read it). The a) and b) labels are set twice.

*Thank you for this comment as well. We corrected the labels and increased the Figure in the revised manuscript.*

**References**

Brugnara, Y., Hari, C., Pfister, L., Valler, V., and Brönnimann, S.: Pre-industrial temperature variability on the Swiss Plateau derived from the instrumental daily series of Bern and Zurich, Climate of the Past Discussions, pp. 1–34, https://doi.org/10.5194/cp-2022-34, 2022.

Hufkens, K., Basler, D., Milliman, T., Melaas, E. K., and Richardson, A. D.: An integrated phenology modelling framework in r, Methods in Ecology and Evolution, 9, 1276–1285, https://doi.org/10.1111/2041-210X.12970, 2018.

Imfeld, N., Pfister, L., Brugnara, Y., and Brönnimann, S.: A 258-year-long data set of temperature and precipitation fields for Switzerland since 1763, Climate of the Past, 19, 703–729, https://doi.org/10.5194/cp-19-703-2023, 2023.

Slivinski, L. C., Compo, G. P., Sardeshmukh, P. D., Whitaker, J., McColl, C., Allan, R., Brohan, P., Yin, X., Smith, C., Spencer, L., , Vose, R. S., Rohrer, M., Conroy, R. P., Schuster, D. C., Kennedy, J. J., Ashcroft, L., Brönnimann, S., Brunet, M., Camuffo, D., Cornes, R., Cram, T. A., Domínguez-Castro, F., Freeman, J. E., Gergis, J., Hawkins, E., Jones, P. D., Kubota, H., Lee, T. C., Lorrey, A. M., Luterbacher, J., Mock, C. J., Przybylak, R. K., Pudmenzky, C., Slonosky, V. C., Tinz, B., Trewin, B., Wang, X. L., Wilkinson, C., Wood, K., and Wyszyński, P.: An evaluation of the performance of the twentieth century reanalysis version 3, Journal of Climate, 34, 1417–1438, https://doi.org/10.1175/JCLI-D-20-0505.1, 2021.

Wang, H., Wu, C., Ciais, P., Peñuelas, J., Dai, J., Fu, Y., and Ge, Q.: Overestimation of the effect of climatic warming on spring phenology due to misrepresentation of chilling, Nature Communications, 11, 4945, 2020.

**Response to Reviewer #3**

This is overall well-written, and the approaches are appropriate, I have provided an annotated copy of the manuscript to assist the authors in making corrections. I have three substantive points below which require addressing.

*Thank you for your detailed review of our manuscript and your constructive comments. In the following, we address the three substantive points raised by you. We are also thankful for the remarks made directly in the manuscript.*

1) The snow days approach presented is not actually documenting snow days – you are presenting 'potential snow days'. What you are presenting is a 'potential snowfall days', as no observation of snowfall are made. This may seem rather pedantic but is important from an accuracy perspective, and the method you apply is likely to result in substantial overestimation of snow days. I recommend rephrasing the snow days to potential or removing this from the paper as it is likely to result in an ill-informed discussion. The sensitivity of such an approach in a country with extensive elevation differences is also liable to result in high uncertainties. I would remove this section from the paper as it I fear detracts from the paper overall, an alternative approach would be to demonstrate that the method you apply provides a good proxy for snow days for the different regions using observational records.

*Thank you for this comment. We use a very simple approach to deduce days with snowfall from daily precipitation and daily mean temperature. Such a simple approach is needed because we only have these two variables available for the period back to 1763. The evaluation of Zubler et al. (2014) for the 2x2 km grid and 46 stations of the National Basic Climatological Network between 1980 - 2009 shows high Spearman correlations of 0.93 and a mean bias of -0.23 days stating, however, that the biases may come from the smoothed 2-km topography. We calculated a similar evaluation for our 1x1 km grid with 27 stations across Switzerland. With a mean Spearman correlation of 0.8 and a mean bias of -0.3, our results are not as good as in Zubler et al. (2014). However, Zubler et al. (2014) evaluated the number of days with snowfall for an entire year, whereas we considered the spring season only. Estimating snowfall days in spring based on such threshold may be more difficult than in e.g. winter which would explain the difference in results. Nevertheless, we will keep the snowfall days index in the manuscript because we think that an estimation of snowfall days in spring is relevant for illustrating changes in spring weather in Switzerland. Also, we think it would be valuable to evaluate the snowfall estimates in more detail. This is, however, beyond the scope of this article. We added the following paragraph to the manuscript describing the evaluation results, and we added that it is a snowfall day estimate in the results and discussion text of the manuscript.*

*"We evaluated these thresholds by comparing the closest grid cells of snowfall days with observations of snowfall days at 27 different stations across Switzerland for the spring season. The results showed a mean Spearman correlation of 0.8 and a mean bias of -0.3 d. Especially for stations above 1000 m, the gridded data set showed a tendency to underestimate the number of snowfall days. It is therefore important to remember that our snowfall days only represent potential snowfall days. Nevertheless, these estimates provide a good basis for evaluating spring snowfall over the historical period."*

2) I am concerned by your use of the period 1871-1900 as pre-industrial, I appreciate that parts of Switzerland may not have been industrial, but certainly neighbouring regions and countries had extensive industry by this stage. I think using any period post-1750/1800 as pre-industrial is fraught with risk, particularly when dealing with climate and climatic parameters which are influenced beyond regional boundaries.

*This period has been suggested by the Swiss national weather service based on evaluations using different periods and taking into consideration comparability to other periods and countries (Begert et al., 2019). Thus, we used the same period for consistency with respect to other publications by the Swiss national weather service. We kept the 1871 to 1900 period as a reference period mentioning that it is used by the national weather service as a pre-industrial reference, but we removed the term pre-industrial from the rest of the manuscript.*

3) The discussion would benefit from a section explaining how the results of this paper relate to studies in neighbouring regions/countries, can you provide a little further context please of how this fits with other studies in Central Europe.

*We extended the discussion by adding three small paragraphs in the discussion section, considering studies for Italy and the Czech Republic, and for a general Central Europe context.*

*Further, we corrected the mistakes marked by you directly in the manuscript. Thanks a lot for the careful review! Also, we decided to change Figure 1 by creating two Figures covering the different indices. The anomalies with*

*respect to the period 1871 to 1900 are provided in the Appendix, but we did not add them to the main manuscript. Also, we decided not to evaluate whether statistical step changes were present around the 1980s, since several studies have been published (e.g. Marty, 2008; Reid et al., 2016) and possible explanations provided (Sippel et al., 2020).*

**References**

Begert, M., Stöckli, R., and Croci-Maspoli, M.: Klimaentwicklung in der Schweiz - vorindustrielle Referenzperiode und Veränderung seit 1864 auf Basis der Temperaturmessung, Technical Report Meteoswiss, 274, 23 pp, URL `https://www.meteoswiss.admin.ch/dam/jcr:4c89a839-d577-47f1-aeb9-a30749ddaf2b/AB_Vorind_Refp_v1.1_de.pdf`, (last accessed 30 June 2023), 2019.

Marty, C.: Regime shift of snow days in Switzerland, Geophysical Research Letters, 35, https://doi.org/10.1029/2008GL033998, 2008.

Reid, P. C., Hari, R. E., Beaugrand, G., Livingstone, D. M., Marty, C., Straile, D., Barichivich, J., Goberville, E., Adrian, R., Aono, Y., Brown, R., Foster, J., Groisman, P., Hélaouët, P., Hsu, H.-H., Kirby, R., Knight, J., Kraberg, A., Li, J., Lo, T.-T., Myneni, R. B., North, R. P., Pounds, J. A., Sparks, T., Stübi, R., Tian, Y., Wiltshire, K. H., Xiao, D., and Zhu, Z.: Global impacts of the 1980s regime shift, Global Change Biology, 22, 682–703, https://doi.org/10.1111/gcb.13106, 2016.

Sippel, S., Fischer, E. M., Scherrer, S. C., Meinshausen, N., and Knutti, R.: Late 1980s abrupt cold season temperature change in Europe consistent with circulation variability and long-term warming, Environmental Research Letters, 15, 094 056, https://doi.org/10.1088/1748-9326/ab86f2, 2020.

Zubler, E. M., Scherrer, S. C., Croci-Maspoli, M., Liniger, M. A., and Appenzeller, C.: Key climate indices in Switzerland; expected changes in a future climate, Climatic change, 123, 255–271, https://doi.org/10.1007/s10584-013-1041-8, 2014.

---

## Author Response (AR2)

Dear Editor,

Thank you for the suggested citations. We added these citations in the discussion section of the manuscript.

Kind regards,
Noemi Imfeld